# Effect of Dietary Incorporation of Hemp Seeds Alone or with Dried Fruit Pomace on Laying Hens’ Performance and on Lipid Composition and Oxidation Status of Egg Yolks

**DOI:** 10.3390/ani14050750

**Published:** 2024-02-28

**Authors:** Daniel Mierlita, Alin Cristian Teușdea, Mădălina Matei, Constantin Pascal, Daniel Simeanu, Ioan Mircea Pop

**Affiliations:** 1Department of Animal Nutrition, Faculty of Environmental Protection, University of Oradea, 1 University St., 410087 Oradea, Romania; ateusdea@gmail.com; 2Faculty of Food and Animal Sciences, “Ion Ionescu de la Brad” University of Life Sciences, 8 Mihail Sadoveanu Alley, 700490 Iași, Romania; madalina.matei@uaiasi.ro (M.M.); pascalc@uaiasi.ro (C.P.); daniel.simeanu@iuls.ro (D.S.)

**Keywords:** dried blackcurrant pomace, dried rosehip pomace, cholesterol, health lipid indices, antioxidants, lipid oxidation

## Abstract

**Simple Summary:**

Developing new products according to consumer wishes is a strategy that most industries use to be competitive in the market. The aim of this study was to improve egg quality by reducing cholesterol and saturated fatty acids (SFAs) and enriching polyunsaturated fatty acids (PUFAs) and natural antioxidants. We evaluated the effects of introducing hemp seeds, as a source of PUFAs, into a standard diet with or without dried fruit pomace (dried blackcurrant (DB) or dried rosehip (DR)), as a source of natural antioxidants, on the laying performance of hens and the fatty acid (FA) profile, cholesterol level, antioxidant content, and lipid oxidative status of the yolks of fresh and stored eggs (refrigerated at 4 °C for 28 days). The dietary incorporation of hemp seeds improved the fat quality of the egg yolks by reducing the concentration of cholesterol and SFAs and increasing the proportion of omega-3 FAs. Laying performance, cholesterol concentration, and yolk FA profile were not improved by the use of dried fruit pomace, compared to the diet supplemented only with hemp seeds. The incorporation of dried fruit pomace into the diets increased the antioxidant content and oxidative stability of fats in the yolks, and it improved the color of the yolks. It can be concluded that including hemp seeds in combination with dried fruit pomace in the diets of laying hens did not affect the laying performance of hens but ensured an improvement in egg quality. This strategy of feeding hens allows one to obtain eggs enriched in omega-3 FAs and natural antioxidants, providing an alternative for consumers to obtain these valuable health-promoting nutrients.

**Abstract:**

The present study was conducted to investigate the effects of introducing hemp seeds, as a source of PUFAs, into a standard diet with or without dried fruit pomace (dried blackcurrant (DB) or dried rosehip (DR)), as a source of natural antioxidants, on the laying performance of hens and the FA profile, cholesterol level, antioxidant content, and lipid oxidative status in the yolks of fresh eggs or eggs stored at 4 °C for 28 days. The experiment used 128 Tetra SL hens at 35 weeks of age, which were divided into four groups and randomly assigned to four dietary treatments: a standard corn–wheat–soybean meal diet (C), standard diet containing 8% ground hemp seed (H), hemp seed diet containing 3% dried blackcurrant pomace (HB), and hemp seed diet containing 3% dried rosehip pomace (HR). The laying rate, feed conversion ratio (FCR), egg weight, and yolk weight were improved by the use of hemp seeds. The yolks of the H, HB, and HR eggs had a lower cholesterol (*p* ˂ 0.01) and SFA content, while the concentration of total and individual PUFAs (n-6 and n-3 FAs) was significantly higher (*p* ˂ 0.01) compared to C. In addition, the introduction of hemp seeds into the diets alone or with dried fruit pomace (DB or DR) led to increased (*p* ˂ 0.001) content of α-linolenic acid (ALA, 18:3n-3), eicosapentaenoic acid (EPA, 20:5n-3), and docosahexaenoic acid (DHA, 22:6n-3) and hypo-/hypercholesterolemic FA ratio and decreased arachidonic acid (AA, 20:4n-6) content, n-6/n-3 ratio, and thrombogenicity index (TI) compared to the control eggs. The introduction of dried fruit pomace (DB or DR) into the diets had no effect on the laying performance of the hens or the cholesterol content and FA profile of the egg yolks, compared to the diet supplemented only with hemp seeds. The dried fruit pomace improved the color, accumulation of antioxidants, and oxidative stability of fats in the yolks of the fresh eggs and eggs stored at 4 °C for 28 days. The DR was found to have the most desirable effects, producing the most intense color of egg yolks, the highest content of natural antioxidants, and the best oxidative stability of yolk lipids.

## 1. Introduction

Eggs are an important source of quality protein for the human diet. However, eggs are high in cholesterol and saturated fatty acids (SFAs), and it is claimed that these contribute to coronary heart disease [1]. For this reason, numerous studies have been carried out on reducing the cholesterol and SFA content of egg yolk and enriching omega-3 FA by manipulating the diets of laying hens [2].

It is well known that omega-3 FAs, especially alpha-linolenic acid (ALA), eicosapentaenoic acid (EPA), and docosahexaenoic acid (DHA), have many benefits for human health, including preventing cardiovascular diseases [3] and improving immune functions and fertility [4], as well as anti-inflammatory, antitumor, and antiviral properties [5]. Eggs contain low levels of n-3 FA, but the FA profile of egg yolk fats could be improved by adding natural sources of n-3 FA to the diets of hens. Previous reports have shown that table eggs can be enriched with n-3 FA by adding flax seeds [2,6,7], hemp seeds [8,9,10,11], or hemp seed cake [11] to the diet.

The significant increase in hemp seed production in European countries [12,13,14] and the valuable nutritional content create opportunities to use hemp seeds as a valuable ingredient in animal feed. Whole hemp seeds contain approximately 25% crude protein (CP), 33–35% fat, 34% carbohydrates, crude fiber, vitamins, minerals, and functional components [8,11,15]. Hemp seed fats are high in PUFAs (78.61% of total FAs), including linoleic acid (54.80%) and α-linolenic acid (18.63%) [15]. The main reason why hemp seeds could be used in hens’ feed is the relatively high level of ALA (18–19%) [15], surpassed only by flax seeds (55–57%) [6]. The introduction of 8% hemp seeds in the diets of laying hens led to enriched omega-3 FAs in egg yolks, specifically ALA, EPA, and DHA [11]. In addition, studies by Neijat et al. [16] proved that introducing hemp seeds into hens’ feed, even at high levels (30%), had no negative effects on egg production, general egg sensory qualities, or bird health.

The disadvantage of enriching eggs with omega-3 FAs is increased susceptibility to lipid peroxidation in the yolks, which would affect the nutritional and sensory quality of the eggs, as well as consumer safety. Increasing the PUFA content in the diets of laying hens by using various oil seeds and vegetable oils was found to increase the incidence of liver hemorrhage [17], probably due to the oxidative rancidity of unsaturated FA. Additionally, eggs enriched in omega-3 FAs have been noted to have a fishy odor, which originates from the oxidation of PUFAs in the yolks [18]. Therefore, it is necessary to enrich eggs with antioxidant compounds (vitamin E, carotenoids, and phenols) along with PUFA enrichment to reduce the oxidation of unsaturated FAs and provide a good source of antioxidants to consumers [19,20].

An important, still little-studied source of natural antioxidants is fruit pomace. Following the processing of fruit by squeezing or crushing to obtain the juice, pomace results as a by-product. Fruit pomace contains parts of the pulp, peel, and seeds and represents 20–40% of the processed fruit mass [21]. Annually, more than 500 million tons of fruit pomace is produced globally [22], for which no sustainable unified management strategy has been developed. Adding these by-products to the diets of laying hens would reduce the need for synthetic antioxidants, which can have a negative impact on the health of both poultry and humans, and it could contribute to enriching the natural antioxidants in eggs, which would have a positive effect on the perception and acceptance of eggs by consumers [23]. This sustainable agri-food system promotes the from-farm-to-fork strategy, which is designed to build a fair, healthy, and environmentally friendly food system. In addition, the utilization of fruit pomace in poultry feed fits perfectly into the contemporary concept of a circular bioeconomy and, at the same time, represents a strategy for environmental protection and the sustainable development of poultry production [21].

Berries, such as blackcurrants and rosehips, are rich in essential fatty acids, polyphenols, tocopherols, carotenes, and vitamin C, hence their high antioxidant capacity. The pulp that remains as a by-product after juicing is still a good source of bioactive compounds (phenols, vitamins, provitamins, and essential fatty acids) [24], and it could be used in the diets of laying hens to naturally improve the nutritional quality of table eggs [22]. In Romania, significant amounts of by-products obtained from the processing of forest fruits result annually. According to INS data (National Institute of Statistics), in 2022 in Romania, over 5000 ha was cultivated with forest fruits (blackberries, currants, blueberries, raspberries, and strawberries), and over 9000 tons of forest fruits was harvested from spontaneous flora and 4000 tons of rosehips [25].

Loetscher et al. [26] reported that adding 25 g/kg of dried and ground rosehip fruit to feed significantly slowed the process of lipid oxidation in the meat of broiler chickens. Similarly, Grigorova et al. [27] found that, by adding rosehips to the feed of laying hens (0.5%), the color of the yolk improved significantly, and the level of malondialdehyde (MDA, a marker of lipid peroxidation) in the yolk decreased significantly during storage in a refrigerator or at room temperature for 30 days. Similar results were reported by Vlaicu et al. [6], who supplemented the diets of laying hens with 30 g/kg of dried rosehips. The antioxidant effects of blackcurrant pomace were demonstrated in a study on turkeys [28].

The limited number of studies that tested the addition of dried fruit pomace to the diets of laying hens [29,30], but especially the lack of studies using dried blackcurrant (DB) or dried rosehip (DR) pomace as a source of natural antioxidants in PUFA-enriched feed for laying hens, prompted the authors to conduct this study. The purpose of this study was to evaluate the effects of including hemp seeds (8%) alone or with dried fruit pomace (DB or DR) (3%) in the diets of layer hens on the hens’ performance, as well as the cholesterol, fatty acid profile, antioxidant content, and lipid oxidative status of the yolks of fresh or stored eggs (refrigerated at 4 °C for 28 days). In this experiment, we evaluated the hypothesis that the simultaneous inclusion of hemp seeds and dried fruit pomace in the diets of laying hens increases the concentration of PUFA and natural antioxidants in the yolk and improves the oxidative stability of the egg yolk without affecting egg production.

## 2. Materials and Methods

### 2.1. Ethical Approval

The experimental protocol was approved by the Ethics Committee of the University of Oradea and complied with the legislative regulations (Law 206/2004, Directive 2010/63/EU, Law 43/2014) regarding the use of animals for scientific purposes.

### 2.2. Experimental Materials

The hemp seeds (Jubileu variety) used in this study came from a culture approved for the production of seeds intended for obtaining commercial vegetable oil. Dried blackcurrant (DB) and dried rosehip (DR) pomace were produced by pressing (Bucher HPX presse; Bücher-Unipektin, Niederweningen, Switzerland) at a commercial fruit processing factory in Zalău (Sălaj county, Romania). Immediately after the pomace was obtained, it was dried in a convection oven at 60 °C and then ground using a universal hammer mill with a 1 mm mesh. The dried fruit pomace was kept under vacuum in dark-colored foil bags until use.

Before being added to the diets of laying hens, the hemp seeds and dried fruit pomace (DB and DR) were analyzed for their proximate composition (dry matter (DM), crude protein (CP), ether extract (EE), neutral detergent fiber (NDF), and acid detergent fiber (ADF)) and antioxidant content (α-tocopherol, β-carotene, and total phenols).

### 2.3. Experimental Design

An 8-week experiment was conducted with 128 Tetra SL laying hens 35 weeks of age (initial body weight: 1694.2 ± 87.63 g), which were purchased from a commercial farm (SC Rosbro Avicom SRL, Bihor, Romania). They were divided into 4 homogeneous groups of 32 hens each (8 replicates/group with 4 hens/replicate). Each group of hens was randomly assigned to 1 of 4 experimental diets (treatments): a standard diet for laying hens based on corn, wheat, and soybean meal (control diet, C); standard diet containing 8% ground hemp seed (H); hemp seed diet containing 3% dried blackcurrant pomace (HB); and hemp seed diet containing 3% dried rosehip pomace (HR). Application doses of 3% for the dried fruit pomace were established in accordance with previous studies that demonstrated that the optimal dose of sources of natural antioxidants (tomato waste, dehydrated carrots, rosehip meal, dehydrated sea buckthorn pomace, and dehydrated kapia peppers) incorporated into the diet of laying hens is 2–3% [6,7]. In addition, Konca et al. [30] reported that rosehips can act as a pro-oxidant at high concentrations of 5% in the diet of laying quail.

The hens were raised in a shelter equipped with Zucami three-tier metallic cages (60 cm width × 60 cm length × 40 cm height) at a density of 4 hens/cage (900 cm^2^/hen) and a controlled microclimate (temperature: 20–22 °C; humidity 65–68%), which allowed food intake and egg production to be recorded separately for each replicate (cage). Access to food and water was provided ad libitum. Before the start of the experiment, the hens were adapted to their cages and experimental diets for a period of 2 weeks.

Feed was offered once a day at 8:00 a.m., separately for each replicate, in an amount limited to 120 g/hen (400 g/replicate), to reduce feed selection behavior so that the hens would consume almost all food provided. Throughout the experiment, the hens had a schedule of 16 h of light and 8 h of darkness.

### 2.4. Dietary Treatments

Appropriate software (HYBRIMIN^®^ Futter 5) was used to formulate the diets (Table 1) according to the feeding requirements of laying hens [31]. In contrast to diet C, experimental diet H included 8% hemp seeds as a source of PUFAs, while an additional 3% dried blackcurrant pomace was present in diet HB and 3% dried rosehip pomace was present in diet HR, as sources of natural antioxidants. All diets were isocaloric and isonitrogenous, containing 17.5% CP and 2750 kcal/kg metabolizable energy (ME). The ME of the diets was adjusted using sunflower oil.

For good homogenization, whole (intact) hemp seeds were first mixed with wheat grains and then ground before formulating the diets [16]. The diets were stored during the experiment in labeled bags in a cool and dry room.

### 2.5. Performance Parameters

Egg production was recorded daily, and egg weight was determined 3 times per week. Recordings were made for each replicate. Egg mass was calculated based on laying rate and egg weight. Feed consumption was measured each week by weighing the feed at the beginning and end of the period.

### 2.6. Egg Sampling

To determine the physical characteristics of the eggs, 16 eggs/treatment (2 eggs/replicate) were collected twice during the experiment (weeks 6 and 8 of the experimental period). After weighing, the eggs were broken, and the weights of their components were determined; based on these, the percentages of white, yolk, and shell were calculated. Eggshells from the same eggs were washed and left at room temperature for 2 days, after which they were weighed using an electronic scale (Mettler-Toledo LLC, Columbus, OH, USA).

In the last week of the experiment, a total of 256 eggs were taken, 64 eggs for each dietary treatment (2 eggs/replicate × 8 replicates × 4 days); 128 of these eggs (32/treatment) were processed and analyzed as fresh eggs, and 128 eggs (32/treatment) were stored in a refrigerator at 4 °C for 28 days. The eggs were broken, and the yolks were separated. The containers with yolk samples were wrapped with aluminum foil to protect them from light and were frozen at −80 °C until the laboratory analysis of the FA profile, antioxidant content (α-tocopherol, retinol, β-carotene, and total phenols), and lipid oxidative status (antioxidant capacity and MDA concentration) of the yolks.

Yolk pigmentation and cholesterol content were determined in 24 eggs per treatment (3 eggs per replicate), recorded at the beginning of the last week of the experimental period. The yolk color was determined using the Roché scale (1–15 points).

### 2.7. Chemical Analyses

#### 2.7.1. Proximate Chemical Composition of Feed

Samples of the hemp seeds, the dried blackcurrant pomace, the dried rosehip pomace, and each experimental diet (150 g) were analyzed for DM following the gravimetric method [33], CP following the Kjeldahl method (N × 6.25, Kjeltec Auto 1030 Analyzer; Foss Tecator AB, Höganäs, Sweden), EE by petroleum ether extraction using method 920.39 (SOXTHERM, C. Gerhardt GmbH, Königswinter, Germany) [33], and NDF and ADF by means of the method described by Van Soest et al. [34] using an ANKOM 220 analyzer (ANKOM Technology, Fairport, NY, USA). All determinations were performed in triplicate.

#### 2.7.2. Yolk Cholesterol Content

The cholesterol in yolk samples was determined using a Perkin-Elmer gas chromatograph (Shelton, MA, USA) according to AOAC [35]. Dried yolk samples (65 °C) were saponified with methanolic KOH solution (50 mL) in a water bath for 1 h. Next, the samples were treated with petroleum ether, concentrated on a rotary evaporator, and brought to neutral pH with distilled water. After removing the petroleum ether, the residue was treated with chloroform (5 mL). Aliquots of 1 μL of the obtained extracts were injected into an HP-5 GC fused silica capillary column (30 m × 0.32 mm ID, 0.1 µm film thickness; J&W GC Columns, Agilent, Santa Clara, CA, USA) and analyzed on a detector with flame ionization (FID). Cholesterol was identified by comparing the peak areas with those obtained from the laboratory standard solution. The cholesterol concentration is expressed as g/100 g yolk.

#### 2.7.3. Feed and Egg Yolk Fatty Acid Analysis

To determine the FA composition of the feed and egg yolk, we used standard fatty acid methyl ester (FAME) gas chromatography techniques [33]. The fat was extracted with petroleum ether and stored at −20 °C in Eppendorf tubes until laboratory analysis. In the first step, the extracted fat (100 mg) was saponified with 2N KOH (100 μL) and hexane (3 mL). After vigorous shaking for 1 min, the mixture was centrifuged for 5 min at 5000 rpm. A chromatographic analysis was performed with a gas chromatograph (Shimadzu GC-2010 Plus, Tokyo, Japan) equipped with FID and an HP-88 column (100 m long, 0.25 mm diameter, and 0.20 μm film thickness). Helium was the carrier gas (2 mL/min), and the split rate was 1:50. The injector temperature was 240 °C. The temperature program for the oven was set at 100 °C for 1 min, then 100 to 170 °C at 6.5 °C/min, 170 to 220 °C for 10 min at 3 °C/min, and finally 230 °C for 5 min. Each FA was identified using external standards (Supelco 37 Component FAME mix; Supelco Inc., Bellefonte, PA, USA) by comparing retention times. The results are expressed as % FA of total FA.

#### 2.7.4. Determination of Antioxidant Compounds

To determine the contents of retinol and α-tocopherol in the feed and yolk samples, the method described in EC Regulation [36] was used, using a high-performance liquid chromatograph (HPLC) equipped with a PDA-UV detector (Finnigan Surveyor Plus, Thermo Scientific, Waltham, MA, USA) at 325 nm for retinol and 292 nm for tocopherol. A HyperSil BDS C18 column with a silica gel size of 250 × 4.6 nm and particle size of 5 µm (Thermo Scientific, Waltham, MA, USA) was used. The mobile phase was methanol–water (96% methanol and 4% ultrapure water) at a flow rate of 1.5 mL/min.

For the determination of carotene, an HPLC equipped with a PDA-UV detector (Finnigan Surveyor Plus, Thermo Scientific, Waltham, MA, USA) at a wavelength of 450 nm and a C18 column (250 × 4.60 nm, particle size 5 µm) (Nucleodur, Macherey-Nagel, Düren, Germany) was used. The mobile phase was 100% acetone at a flow rate of 0.8 mL/min.

Total phenolic content (TPC) was measured spectrophotometrically according to the Folin–Ciocalteu method as described by Velioglu et al. [37]. Briefly, samples of methanolic extract (0.1 mL) were mixed with Folin–Ciocalteu reagent (0.1 mL) and distilled water (0.8 mL), and then they were homogenized and incubated for 5 min at room temperature. Next, 0.5 mL sodium carbonate solution (20%) was added and incubated at room temperature for 30 min. The samples were stored for 1 h in the dark, and then the absorbance at 750 nm was measured. Gallic acid was used as a standard to obtain a calibration curve, and the results are expressed in milligrams per gallic acid equivalent (mg GAE/g).

#### 2.7.5. Lipid Oxidative Status of the Yolk

The antioxidant activity was determined by the 2,2′-azino-bis(3-ethylbenzothiazoline-6-sulfonic acid) (ABTS) method, as described by Gaffney et al. [38]. Briefly, the ABTS radical cation (ABTS^•+^) was obtained by mixing ABTS solution (7.0 mmol/L) with potassium persulfate (2.45 mmol/L) at a ratio of 1:1 (ABTS:potassium persulfate). The mixture was kept in the dark at room temperature for 15 h and diluted with ethanol to an absorbance of 0.7 ± 0.05. The homogenized yolk samples were treated with the prepared mixture, and, after 30 min of storage at room temperature, the absorbance of the mixture was measured at 734 nm. The results are expressed as μmol Trolox equivalent (TE)/g egg yolk.

The oxidative stability of the yolk was evaluated based on thiobarbituric acid reactive substances (TBARSs), using the method described by Mierlita [11]. Briefly, egg yolk was mixed with trichloroacetic acid and centrifuged at 5500× *g* at 4 °C for 15 min. The supernatant was mixed with a thiobarbituric acid solution (pH 2.5), after which the tubes were placed in a water bath (90 °C) for 30 min. After cooling, distilled water was added, and the mixture was centrifuged again. The colored product formed by the reaction of lipid peroxidation products with thiobarbituric acid was measured spectrophotometrically at 534 nm. To calculate the concentration of malondialdehyde (µg MDA/g yolk), the values obtained were compared with the standard curve prepared by using standard MDA tetrabutylammonium salt (Sigma-Aldrich, Buchs, Switzerland).

### 2.8. Estimation of Health-Related Lipid Quality Indices

Based on the fatty acid composition of the egg yolk, health indices and fatty acid metabolism indices were calculated using appropriate equations, as they were previously validated in other reports [2,6,15,39].

### 2.9. Statistical Analysis

A one-way analysis of variance (ANOVA) using PROC GLM in SAS [40] for a completely randomized design was performed to determine the effects of the dietary treatments on the performance of the laying hens and the quality traits, FA profile, antioxidant content, and lipid oxidative status of the egg yolks. The data on the laying performance of the hens and the quality of the eggs were tested only for the effect of diet, and the other data were tested for the type of diet (C, H, HB, or HR) and type of eggs (fresh or stored). The significance between individual means was identified using Tukey’s multiple range test. The final data are provided as mean values ± standard error of the mean (SEM), with a significance level of *p* < 0.05.

Correlations between variables and the Kruskal–Wallis (*p* = 0.05) non-parametric test followed by multiple pairwise comparisons (Dunn test, *p* = 0.05) were calculated with Stata 17.0 SE Standard Edition (StataCorp LLC, StataCorp, 4905 Lakeway Drive, College Station, TX, USA). A principal component analysis (PCA) was computed with a custom-made program developed in MATLAB 2023a 9.14.0 CWL (The MathWorks Inc., 1 Apple Hill Drive, Natick, MA, USA).

## 3. Results

### 3.1. Chemical Compositions of Hemp Seed, Dried Fruit Pomace, and Experimental Diets

The hemp seeds had higher concentrations of CP and EE (*p* ˂ 0.001), as well as a higher concentration of NDF (*p* ˂ 0.05), than the dried fruit pomace (Table 2). The dried blackcurrant (DB) and dried rosehip (DR) pomace had a higher β-carotene and total phenol content (DR ˃ DB) than the hemp seeds (*p* ˂ 0.001). The α-tocopherol content was higher in the hemp seeds and DR than in the DB (*p* ˂ 0.001). Thus, the DR was the richest source of α-tocopherol, β-carotene, and total phenols. Compared to the DB, the DR had a 3.34 times higher content of α-tocopherol, 1.93 times higher content of β-carotene, and 26.65% higher content of total phenols.

The FA profile a showed higher concentration of oleic acid (OA, C18:1 c9) (*p* ˂ 0.01) and a smaller n-6/n-3 ratio (*p* ˂ 0.05) in the dried fruit pomace than in the hemp seeds (Table 3). The hemp seeds had higher concentrations of LA (C18:2n-6) (*p* ˂ 0.01) and ALA (C18:3n-3) (*p* ˂ 0.05) than the dried fruit pomace. The FA profile of the experimental diets showed a higher concentration of palmitic acid (PA), OA, and LA in the control diet (C) (*p* ˂ 0.05) and a more than 6 times higher concentration of ALA in the experimental diets (H, HB, and HR) (*p* ˂ 0.001), so the n-6/n-3 ratio decreased dramatically, from 10.96 for diet C to 2.05–2.09 for the diets supplemented with hemp seeds alone or in combination with dried fruit pomace (*p* ˂ 0.001) (Table 3).

### 3.2. Performance of Laying Hens and Egg Quality

The body weight (BW) of the laying hens was not affected by the experimental diets (Table 4).

No significant effect was found for daily feed intake, but the laying rate, egg mass, egg weight, and yolk weight increased (*p* ˂ 0.05) in the experimental groups (H, HB, and HR) compared to the control group (C) (Table 4). The feed conversion ratio (FCR) was more favorable (*p* ˂ 0.05) in the groups given feed with hemp seeds with or without dried fruit pomace (H, HB, and HR) than in group C. The weight of the egg albumen and shell was not influenced (*p* > 0.05) by the experimental diets.

Including dried fruit pomace (DB and DR) in the diets of the laying hens did not produce differences (*p* ˃ 0.05) in terms of the feed intake, laying rate, FCR, average weight of eggs, or egg components (albumen, yolk, and shell). However, the presence of dried fruit pomace in the diet improved (*p* ˂ 0.001) the color of the yolk; the Roche yolk color fan (RYCF) score was higher (*p* ˂ 0.001) compared to the control group (C) or the group with the diet that contained only hemp seeds (H). The mean values of the yolk pigmentation scores, determined by the Yolk Color Fan^®^ scale, were higher for HR than for HB (11.32 vs. 9.87; *p* ˂ 0.05) (Table 4).

### 3.3. Egg Yolk Fatty Acid Profile and Cholesterol Content

The most abundant FAs in the egg yolk were PA (C16:0), OA (C18:1 c-9), and LA (C18:2n-6) in all groups of hens (Table 5).

Palmitic and stearic acids were the most abundant saturated FAs in the eggs from group C, leading to higher total SFAs (*p* ˂ 0.05) compared to in H, HB, and HR. The OA content was lower (*p* ˂ 0.01) in the eggs from the hens fed the diet containing hemp seed alone or in combination with dried fruit pomace (H, HB, and HR), leading to lower total MUFAs (*p* ˂ 0.01) compared to in C.

The most significant change in the FA profile of the yolks was observed for the total n-3 FA concentration, which was more than threefold higher (*p* ˂ 0.001) in the H, HB, and HR eggs compared to in the C eggs, leading to an increase (*p* ˂ 0.001) in total PUFAs in the H, HB, and HR eggs. In addition, the HR and HB eggs had a higher content of n-3 FAs and total PUFAs than H, but without significant differences (*p* ˃ 0.05) (Table 5). Among the n-3 FAs, there were higher concentrations (*p* ˂ 0.001) of ALA (C18:3n-3) and long-chain n-3 FAs (eicosapentaenoic acid (EPA, C22:5n-3) and docosahexaenoic acid (DHA), C22:6n-3)) in the yolks of the H, HB, and HR eggs compared to in the C eggs. The same was the case for LA (C18:2n-6), the best represented n-6 FA (Figure 1). In addition, arachidonic acid (AA, C20:4n-6) was decreased (*p* < 0.05) in the yolks of the eggs from the hens that were given diets containing hemp seeds (H, HB, and HR) compared to in the yolks of the eggs from the control group.

Taking into account the lack of any difference in the FA profile of yolk fats between groups H, HB, and HR, we can assume that the two types of dried fruit pomace (DB and DR) introduced into the diets of the laying hens did not significantly modify (*p* ˃ 0.05) the composition of fatty acids in the egg yolks (Table 5).

Diet had a significant effect on the FA profile of the eggs stored at 4 °C for 28 days; the most important change was found in PUFA concentration. During storage, the concentration of n-3 FA decreased (*p* ˂ 0.05) in the eggs from all groups, while the content of n-6 FA and total PUFAs decreased only in the C, H, and HB eggs (Table 6). In addition, an increase (*p* ˂ 0.01) in SFA concentration in the yolks was observed during storage, but the differences were significant only in the case of the C eggs.

The results regarding the concentration of cholesterol in the egg yolks are presented in Figure 2. Although the fat content of the yolks was not affected by the experimental diets (Table 5), the addition of hemp seeds to the diets of the laying hens caused a decrease (*p* ˂ 0.01) in the cholesterol level in the yolks of the eggs from all three experimental groups (1.739, 1.718, and 1.691 g/100 g yolk) compared to in those from group C (1.978 g/100 g yolk).

### 3.4. Egg Yolk Lipid Quality Indices

The most significant alteration in the health-related lipid quality indices was observed in the fatty acid ratios n-6/n-3 and LA/ALA, with both being lower (*p* < 0.001) in the H, HB, and HR eggs compared to in C (Table 7). The n-6/n-3 ratio in the egg yolks was 4.98 ± 0.24 (H), 4.79 ± 0.19 (HB), and 4.73 ± 0.25 (HR), i.e., three times lower (*p* < 0.001) than in the C eggs (14.10 ± 0.23).

The incorporation of hemp seeds into the diets of laying hens (H, HB, and HR) did not affect the nutritional value index (NVI) or atherogenicity index (AI) but reduced (*p* ˂ 0.001) the thrombogenicity index (TI) and increased the health-promoting index (HPI) and desirable FA (DFA), as well as the hypocholesterolemic/hypercholesterolemic (h/H) FA ratio (Table 7). As expected, the peroxidability index (PoI) and oxidative susceptibility (OS) were higher (*p* ˂ 0.001) in groups H, HB, and HR compared to in group C. Among the FA metabolism indices, only the elongase index (EI), the thioesterase index (TsI), and ∆5/∆6-desaturase were influenced by the dietary treatments. The addition of hemp seeds alone or with dried fruit pomace in the diets of laying hens (H, HB, and HR) caused an increase (*p* ˂ 0.01) in EI and TsI and a decrease (*p* ˂ 0.01) in ∆5/∆6-desaturase.

### 3.5. Antioxidant Compounds and Lipid Oxidative Status of Yolks

The concentrations of the antioxidants (α-tocopherol, retinol, β-carotene, and total phenols) in the yolks before and after storage at 4 °C for 28 days are reported in Table 8. The concentration of α-tocopherol increased (*p* ˂ 0.001) in fresh eggs from the hens fed the experimental diets (H, HB, and HR), while the yolks of the eggs from the hens fed diets enriched with dried fruit pomace (DB or DR) contained higher amounts of retinol (by 17.48–27.6%; *p* < 0.05), β-carotene (by 1.1–1.6 times; *p* < 0.001), and total phenols (by 40.9–41.7%; *p* < 0.05) compared to those from group C. The storage of eggs at 4 °C for 28 days led to significant reductions in the α-tocopherol, retinol, and total phenol content in the yolks of the eggs from all groups (*p* ˂ 0.05) (Table 8).

In this study, differences in the egg yolk antioxidant content corresponded to differences in antioxidant activity. The highest antioxidant activity was observed in the eggs from the hens whose diet contain dried fruit pomace (HB and HR), in both fresh eggs (1.273–1.281 μmol TE/g yolk; *p* ˂ 0.01) and stored eggs (0.479–0.495 μmol TE/g yolk; *p* ˂ 0.05) (Table 8). The antioxidant capacity was higher for the H eggs than for the C eggs, proportional to the higher concentration of α-tocopherol in the yolks. The antioxidant capacity was 2.6–2.8 times lower in the stored eggs than in the fresh eggs (*p* ˂ 0.001).

Increasing the concentration of PUFAs by incorporating hemp seeds into the diet (H, HB, and HR) did not have an effect on reducing the oxidative stability of the lipids in the yolks of the fresh and stored eggs, with lower MDA concentrations than in the eggs from the hens fed the standard diet (C). The MDA content, which is a marker of lipid peroxidation, increased significantly during storage in the eggs from all groups (*p* ˂ 0.001) (Table 8). Thus, after 28 days of refrigerated storage, the MDA content in the yolks (µg MDA/g) increased by 71.2% in the eggs from group C and by 75.1%, 77.0%, and 60.2% in the eggs from groups H, HB, and HR, respectively; despite being enriched with PUFAs, they are much more susceptible to lipid oxidation. The TBARS values decreased (*p* < 0.01) in both the fresh and stored eggs from the groups supplemented with DB or DR, with the dried rosehip pomace presenting the best efficiency (*p* < 0.001) in slowing down lipid degradation.

Pearson’s correlation showed that n-3 FA was positively correlated with the total PUFA (r = 0.925, *p* < 0.001) and α-tocopherol contents (r = 0.671, *p* < 0.001) and negatively with cholesterol concentration (r = 0.570, *p* < 0.001) and MDA (r = 0.922, *p* < 0.001) in the stored egg yolk. There was a strong positive correlation between antioxidant capacity (TAC) and the α-tocopherol (r = 0.404, *p* ˂ 0.05), retinol (r = 0.660, *p* < 0.001), β-carotene (r = 0.406, *p* ˂ 0.001), and total phenol (r = 0.604, *p* < 0.001) concentration in the yolks of the eggs from all groups (Figure 3). The concentration of MDA in the yolk was strongly negatively correlated with its content in antioxidants (retinol, total phenols, β-carotene, and α-tocopherol; *p* ˂ 0.01).

The FA and antioxidant content and lipid oxidative status of the yolk of the stored eggs were analyzed by PCA (principal component analysis), which allows for the identification of relationships between explanatory variables and a good separation of groups, thus reducing the complexity of multivariate data (Figure 4). The first principal component, PC1 (eigenvalue: 8.41), explained most of the data variability (64.68%), and the second component, PC2 (eigenvalue: 1.45), explained less of the variance (11.16%). The first two principal components (PC1 and PC2) explained 75.85% of the variability, while PC1 and PC3 presented a value of 71.35%. The H eggs had the highest concentration of PUFA (n-3 FA and n-6 FA) and α-tocopherol, and the use of dried fruit pomace in the hens’ diet caused an increase in the content of antioxidants (retinol, β-carotene, and total phenols) and antioxidant activity of the egg yolk, while the C eggs had a higher cholesterol, SFA, and MDA content. The loadings for PC3 split the H eggs, showing the positive effects of hemp seed dietary supplementation on cholesterol, MDA, and α-tocopherol concentration and egg yolk antioxidant activity, and the negative effects on SFA content and n-6/n-3 FA.

## 4. Discussion

### 4.1. Chemical Composition of Hemp Seed and Dried Fruit Pomace

As expected, the hemp seeds had higher protein and fat concentrations (*p* ˂ 0.001) than the dried fruit pomaces (DB and DR), which instead were a good source of β-carotene and total phenols (*p* ˂ 0.001). The higher concentration of α-tocopherol in the DR compared to in the DB is attributed to the seed content, which also provided a higher fat content for the DR. Previously, Vlaicu et al. [6] reported that rosehip seeds have high contents of oil (16.2%) and α-tocopherol. 

Both the hemp seeds and dried fruit pomace were good sources of n-3 FA and n-6 FA, with a similar n-6/n-3 ratio (Table 3). However, the chemical composition of dried fruit pomace is influenced by a number of factors, such as the fruit processing method, climatic factors or country of origin [6], harvest period, soil type, and cultivation technology [27]. Similarly, the chemical composition of hemp seeds varies depending on different parameters such as environmental conditions, soil composition, fertilizers, and plant variety [14]. In our research, the DB had a total phenol content of 19.32 mg GAE/g. Previous studies reported total phenol concentrations of 6.0 mg GAE/g [41] and 31.0 mg GAE/g [28] for DB. These differences can be attributed to the content of anthocyanins, which, depending on the method of juice production, can pass into the juice and, thus, are present in variable amounts in the pomace [42]. The total phenol content of the hemp seeds is lower than that reported in other studies (6.41 vs. 20.19 mg GAE/g) [14]. This difference could be attributed to plant variety (different genotypes) and environmental conditions (year of growth) [12]. Moreover, it must be taken into account that phenols present a very small portion of the polyphenolic profile, and the evaluation of phenols by the Folin–Ciocalteu method always presents lower values than methods that use different solvent mixtures [37].

### 4.2. Performance of Laying Hens and Egg Quality

In the present study, feed intake was not affected by dietary treatments. However, laying rates were high (*p* ˂ 0.05), and egg weight and yolk weight increased, which allowed for a better feed conversion rate (FCR) for the H, HB, and HR diets than for the C diet. Similar results have been reported previously [8,11,13,16,43], demonstrating that adding up to 30% hemp seeds to the diets of laying hens increased egg weight and egg mass. However, Neijat et al. [16] reported that adding 10% hemp seeds to the diets of laying hens did not affect the laying rate but reduced the egg weight. Also, our findings contrast with those of other studies that found that including 8% and 12% hemp seeds in the diets of laying hens had a negative impact on egg production and feed efficiency (FCR) [44]. These differences may be due to the fact that a newly developed variety of hemp seeds in Turkey was used in the previous study, which may have a higher content of anti-nutrients, exceeding the tolerable limits of laying hens.

The addition of hemp seeds did not affect the weight or the proportion of egg white, which agrees with the conclusions of Skřivan et al. [43] but not with those of Halle and Schone [45], who observed a higher proportion of egg white. In the present study, the experimental diets did not affect eggshell weight, contrary to the results reported by Konca et al. [13], who found an increase in eggshell weight when the diets of laying hens were supplemented with 15% hemp seed. According to Farinon et al. [46], hemp seeds are high in Ca, P, and Mg, which are major components of eggshell, but the amount of hemp seeds used in the present study was probably too small to support the increase in the mineral weight of the eggshells.

Similar to this study, other studies [6,26,27,47] concluded that supplementing the diet with up to 3% rosehips did not affect the BW; daily feed intake; laying rate of hens; egg weight; or egg quality parameters, such as yolk weight, eggshell weight, and eggshell thickness. However, other authors [48] reported that 10% to 15% rosehip meal introduced into the diet caused increased feed consumption and had a negative effect on egg production, probably due to the high content of lignified cell walls and tannins, which reduce the digestibility of nutrients in feed [49]. At the same time, a study by Kaya et al. [48] demonstrated that adding 15% rosehip seeds to the diets of laying hens resulted in eggs with a higher yolk and shell weight and a greater shell thickness, and a more intense color of the yolks due to the high Ca and carotenoid content in the rosehip seeds.

Similar to the present study, Sosnowka-Czajka and Skomorucha [50] did not observe any effect of diets with blackcurrant pomace (3%) on the laying rate, egg weight, or FCR; instead, they found an improvement in the immunological status of laying hens. In turn, Loetscher et al. [26] observed no effect of including dried black chokeberry pomace (2.5%) in the diets of laying hens on egg production, egg weight, feed intake, or feed conversion.

The yolks of the eggs from the hens fed diets containing dried fruit pomace (HB and HR) were found to be more intensely colored (higher RYCF score) than yolks of the eggs from the control group (Table 4). This color improvement is caused by the carotenoids, xanthophylls, and chlorophylls in the pomace, which are transferred to the yolks [44]. In the present study, the color score was higher for the HR eggs than for the HB eggs. Improvements in egg yolk color when the diets of hens were supplemented with rosehip meal were previously reported by Vlaicu et al. [6] and were attributed to the high content of carotenoid pigments in rosehips. Many consumers prefer eggs with a more intensely pigmented yolk because they associate the intense color with high contents of nutrients, especially xanthophylls and lutein, which play a role in the prevention of certain eye diseases [51].

### 4.3. Egg Yolk Fatty Acid Profile

The composition of FAs in the eggs was closely related to their presence in the feed. As expected, since the feed for groups H, HB, and HR had more than 4 times the content of n-3 FA, the eggs from these groups had more than 3 times the content of n-3 FA compared to those from group C.

Hemp seeds are among the richest sources of ALA in poultry diets [11], and they may be responsible for the high ALA content in yolks. A close correlation between the ALA content of the diets of laying hens and egg yolks was previously reported by other authors [52]. Meanwhile, new dietary formulations provide a significantly increased content of EPA (C20:5n-3) and DHA (C22:6n-3), which play important roles in human health by reducing the risk of chronic diseases [53]. An increased intake of ALA, provided by hemp seeds in feed, ensures increased ALA in the yolks, as well as DHA, resulting from the conversion of ALA. A higher concentration of DHA than EPA in the yolks can be attributed to the conversion of a large amount of EPA to DHA in the liver of hens [2]. A similar phenomenon was also observed by Vlaicu et al. [6] in laying hens given feed with linseed (7%). Similarly, Goldberg et al. [54] reported a significant increase in the amounts of ALA, EPA, and DHA in egg yolk when the diets of hens contained hemp oil.

The introduction of hemp seeds into the diets of hens in this study caused a decrease (*p* ˂ 0.05) in the concentration of AA (C20:4n-6) in the egg yolks. A similar decrease in AA was reported by Mazalli et al. [55], who supplemented the diets of hens with linseed oil, which has a high level of ALA. The authors explained that a high level of ALA in the diet decreased AA synthesis from LA (as a precursor) because ALA competes with LA for the same enzyme (∆6 desaturase) that converts ALA to EPA and LA to AA. In our study, hemp seeds caused a decreased AA content in the yolks, probably through the same mechanism as suggested by Mazalli et al. [55]. In addition, the present study also found that the increased ALA level in the diets that contained hemp seed was accompanied by an increased (*p* ˂ 0.001) content of EPA in the yolk lipids, probably due to the increased activity of desaturase used in the conversion of ALA to EPA [56]. Also, Goldberg et al. [54] reported that a reduced LA:ALA ratio favors DHA and EPA synthesis.

Based on the fatty acid profile of the yolk lipids, we calculated that the eggs obtained from the hens fed the experimental diets (H, HB, and HR) would provide approximately 136.0 mg of ALA, 16.60 mg of EPA, and 78.70 mg of DHA. Considering that the recommended daily intake is 1.1–1.4 g of ALA and 300–500 mg of EPA and DHA [3], it follows that the eggs obtained in our study would provide approximately 10% of the daily requirement of ALA and 19 to 26% of the daily requirement of EPA and DHA. Eggs such as those obtained in this study could be the main source of EPA and DHA, especially for people who do not consume fish, since these two long-chain n-3 FAs are mainly present in fish products [2]. However, eggs obtained from hens given feed supplemented with flax seed (4.5% in the feed) had a higher EPA and DHA content, providing approximately 28 to 46% of the recommended daily intake [2]. This difference can be attributed to the higher ALA content of flax seeds (over 50% of total FAs) compared to hemp seeds (19.23% of total FAs) (Table 3).

Limited data are available in the studied literature on the effects of antioxidants in the diets of laying hens on preserving the quality of PUFA-enriched eggs during storage, as more attention has been paid to the quality of freshly laid eggs. The results obtained in this study show that the dietary treatments did not affect the concentrations of SFAs or MUFAs in the egg yolks after 28 days of storage at 4 °C. Contrary to our results, Hayat et al. [19] reported higher levels of MUFAs in the yolks of stored eggs obtained from hens given feed supplemented with flax seed and antioxidants. After 28 days of storage, the total n-3 FA content was reduced by 5.7–9.7% in the eggs from the groups given feed supplemented with hemp seeds and dried fruit pomace (HB and HR) (*p* ˂ 0.05) and by 16.3% in the eggs from the group given feed supplemented only with hemp seeds (H) (*p* ˂ 0.01). The results of the current study are supported by those of the research by Hayat et al. [19], who observed a smaller reduction in the total n-3 FA content after 20 days of storage in eggs from groups given feed supplemented with flax and antioxidants compared to flax alone, demonstrating the importance of antioxidants in protecting PUFA from degradation by oxidation.

### 4.4. Egg Yolk Cholesterol Content

Due to their high cholesterol content, it is thought that eggs have adverse effects on human health [57]. The results obtained in the present study show that, by adding hemp seeds to the diets of laying hens, the concentration of cholesterol in egg yolks can be reduced. Shahid et al. [52] and Golimowski et al. [58] attributed the decreased cholesterol levels in yolks to the content of *β*-sitosterol (phytosterol) in hemp seeds, which plays a role in reducing cholesterol absorption through crystallization and coprecipitation. Moreover, *β*-sitosterol has lower hydrosolubility than cholesterol, so it misplaces the cholesterol from intestinal micelles. This competition reduces the absorption rate of cholesterol [52]. Moreover, phytosterols can decrease hepatic cholesterol biosynthesis, limiting the amount of cholesterol in yolk [58]. Similar findings were previously reported by Johansson [59], who concluded that adding hemp seeds to the feed of laying hens led to a decreased cholesterol content in egg yolks due to the high phytosterol and PUFA content. Vlaicu and Panaite [60] demonstrated the effectiveness of pumpkin seed sterols in reducing egg yolk cholesterol. In agreement with the conclusions of our study, the data available in the specialized literature show that supplementing the diets of laying hens with flax seeds [61,62], hemp seeds [11], and rapeseed [63] led to a decrease in cholesterol concentration in yolks. However, other studies reported that the fat source in the diets of hens does not influence the level of cholesterol in egg yolks [64].

The incorporation of hemp seed or dried fruit pomace into the diets of laying hens did not cause decreased egg yolk cholesterol levels compared to the diet supplemented only with hemp seeds, according to the results reported by Grigorova et al. [27], who concluded that dietary supplementation with 0.5% dried and milled fruits of rosehip did not change the cholesterol concentration in yolk.

### 4.5. Egg Yolk Lipid Quality Indices

The increase in the n-3 FA content in the yolks of the H, HB, and HR eggs resulted in a decrease in the n-6/n-3 ratio compared to the C eggs (Table 7). Reducing n-6/n-3 and LA/ALA ratios is of interest to human health, because it reduces the risk of chronic diseases, such as cardiovascular disease, obesity, and certain forms of cancer, and it improves the activity of the nervous system [1]. The PUFA/SFA and n-6/n-3 FA ratios are commonly used to assess the effects of the nutritional value of animal fat on consumer health. In general, a ratio of PUFA to SFA above 0.45 and a ratio of n-6/n-3 FA below 4.0 are required in human diets to combat ‘lifestyle diseases’, such as coronary heart disease and cancer [1]. In the present study, the PUFA/SFA ratios (0.59–0.80:1) were higher than the recommended values for all groups, whereas the n-6/n-3 FA ratios were closer to the recommended levels for the hemp seed supplemented groups (4.73–4.98:1).

For human health, higher values of the hypocholesterolemic/hypercholesterolemic (h/H) FA ratio are considered more beneficial due to the effects that hypocholesterolemic FAs (C18:1 and PUFAs) have on cholesterol metabolism [1]. In the present study, due to the increased PUFA concentration in the yolks, the h/H FA ratio increased significantly in the H, HB, and HR eggs compared to in the C eggs, thus making them safer for human health. The same effect was observed for the health-promoting index (HPI).

The atherogenic index (AI), which describes the relationship between pro-atherogenic (saturated) FAs and anti-atherogenic (unsaturated) FAs, and the thrombogenicity index (TI), which evaluates the tendency of FAs to form clots in blood vessels, must have values below 0.5 and 1, respectively, for the FAs to benefit human health [1]. In the present study, the experimental diets caused a decrease (*p* ˂ 0.001) in the TI value in the H, HB, and HR eggs, which fell within the recommended range (0.77–0.79). At the same time, the experimental diets did not affect the AI value, which was at the recommended upper limit (0.54–0.59) in all eggs from all groups.

The peroxidability index (PoI), which shows the relationship between PUFA content and oxidation protection potential [6], had higher values in the H, HB, and HR eggs than in the C eggs (Table 7). These values indicate a higher risk of autoxidation in the H, HB, and HR eggs, because these eggs contain three times more n-3 FAs, which are highly susceptible to oxidation. For desirable fatty acids (DFAs), which represent the sum of anti-atherogenic FAs with a plasma cholesterol-lowering effect, higher values are preferred [1]. In the present study, higher values (*p* ˂ 0.01) for DFAs were found in the yolks of the H, HB, and HR eggs due to the higher content of PUFAs, which have anti-atherogenic effects.

Similar to the present study, other researchers [8,10,11,13,16,54] reported improvements in egg yolk lipid quality indices when the diets of laying hens contained hemp seeds or hemp oil.

From the fatty acid metabolism group of indices, elongase was significantly higher (*p* = 0.004) in the H, HB and HR eggs compared with in C, while thioesterase was significantly higher (*p* = 0.009) in HR compared with in C, H and HB. Similar results were obtained by Vlaicu et al. [6] when the hens’ diet was supplemented with flax seed meal in combination with rosehip meal. It should be noted that these indices have a low relevance for discriminating egg quality because they are not able to discriminate metabolic changes due to dietary effects [14]. No effect was found for ∆9-desaturase, which catalyzes the conversion of C16:0 and C18:0 to C16:1 and C18:1, suggesting that the different concentrations of C16:1 and C18:1 in the eggs were not due to ∆9-desaturase activity.

The ∆5/∆6-desaturase complex represents the most valid tool to verify the capacity of animals to synthesize LC-PUFAs (long-chain PUFAs) from precursors. The lower ∆5/∆6-desaturase index in the H, HB and HR eggs than in C (*p* ˂ 0.01) demonstrates competition between n-6 and n-3 FA substrates in the desaturation and elongation pathway. In agreement with the results of this study, Mazalli et al. [55] demonstrated that diets containing higher amounts of ALA (alpha-linolenic acid) increase the activity of ∆6-desaturase and increase long-chain 22-carbon fatty acids, which may be attributed to the use of ALA as the preferred substrate over LA.

### 4.6. Lipid Oxidative Status of Yolks

Numerous previous studies [6,7,27,60,65] demonstrated that supplementing the diets of laying hens with different sources of natural antioxidants improved egg quality and the antioxidant capacity of eggs. The by-products obtained from the processing of fruits and vegetables contain numerous antioxidants, which can be transferred to eggs. Thus, this study aimed to investigate whether the natural antioxidant content of dried blackcurrant (DB) and dried rosehip (DR) pomace could be transferred to eggs and thus improve the antioxidant content and oxidative stability of eggs naturally enriched in PUFAs.

The results show that the BD and DR had a significant effect on the content of retinol, β-carotene, and total phenols in the egg yolk, which can delay the lipid oxidation processes in the yolk. This conclusion is supported by the lower concentration (*p* ˂ 0.01) of MDA, an end product of PUFA oxidation, in the yolks of the HB and HR eggs than in the H and C eggs (Table 8). In addition, the lower concentration of MDA in the yolks of the HR eggs than in the yolks of the HB eggs can be attributed to the higher content of α-tocopherol in the HR eggs, which reduced the oxidative process of lipids in the yolk. Although, to the best of our knowledge, there have been no previous studies using DB or DR as a source of natural antioxidants in the diets of laying hens supplemented with PUFA-rich fats, our results are comparable to those of studies that used rosehip meal, buckthorn pomace, dried carrot, dried kapia pepper, dried tomato waste, or grape pomace [6,7,27,66,67] as a source of natural antioxidants for PUFA-enriched eggs.

The antioxidant capacity was higher for the H eggs than for the C eggs, proportional to the higher concentration of α-tocopherol in the yolks provided by the hemp seeds. Other authors reported a high correlation between the antioxidant content of the diets of laying hens, the concentration of antioxidants in egg yolks, and the antioxidant activity [6,7,23,43,68].

In the present study, the antioxidant capacity, which was assessed based on the ABTS method, decreased in the eggs from all groups during storage, which was probably caused by the degradation of natural antioxidants or the formation of new compounds with pro-oxidant activity [69].

Increasing the concentration of unsaturated FAs in eggs favors lipid oxidation. In this study, although the eggs obtained from the experimental groups (H, HB, and HR) were enriched in PUFAs, there was a lower concentration of malondialdehyde (MDA), a degradation product of PUFAs, than in group C, in both fresh eggs and eggs stored for 28 days in a refrigerator. This finding is difficult to compare with that of other studies because there is a lack of similar work in the literature. The eggs collected from the HR group (hemp seed + DR) had the lowest MDA, in both fresh and stored eggs, which may be because the dried rosehip pomace had the highest antioxidant content. Other previous studies reported an inverse relationship between the content of antioxidants and the concentration of MDA in egg yolk [7,14].

The results of the current study are supported by the findings of Grigorova et al. [27], who reported a significant increase in pigmentation and reduction in MDA levels in egg yolks during 30-day refrigerated storage when the diets of laying hens were supplemented with 0.5% dried and ground rosehips. Similarly, other studies [2,6,7,47] reported that supplementing the diets of laying hens with dried and ground rosehips, sea buckthorn pomace, or tomato peels had a significant influence on lipid peroxidation in eggs during storage. Notably, in this study, the DR was more effective than the DB in reducing the MDA concentration in the yolks of the eggs stored for 28 days in a refrigerator. This superior effect of the DR could be related to the greater amount of antioxidants (Table 2), as well as the presence of additional antioxidants, such as lycopene and vitamin C [6,27,68], which potentiates antioxidant properties. Although it had a lower concentration of antioxidants, the DB was as effective as the DR in decreasing the concentration of MDA in the egg yolks. This effect could also be attributed to the presence of additional natural antioxidants in dried blackcurrant pomace, such as anthocyanins [24], which potentiate antioxidant activity.

## 5. Conclusions

The results of this study provide clear evidence demonstrating that it is safe to add hemp seeds (8%) to the diets of laying hens and that they can serve as a source of PUFAs for the production of high-quality table eggs, enriched with omega-3 FAs and a lower cholesterol and SFA content. In addition, a significant enrichment of eggs with α-tocopherol was found, which ensured an improvement in the antioxidant status of the fats in the egg yolk. The addition of 3% dried fruit pomace (DB or DR), which is enriched in PUFAs, to the diets of laying hens can have a positive influence on the parameters of egg quality by increasing the pigmentation and antioxidant content and reducing the MDA concentration in the yolks of fresh or stored eggs (28 days in the refrigerator), demonstrating its efficiency in reducing the oxidation of unsaturated FAs in egg yolks. It can be concluded that including hemp seeds in combination with dried fruit pomace in the diets of laying hens did not affect the laying performance of the hens but ensured an improvement in egg quality. This strategy of feeding hens allows one to obtain eggs with a lower cholesterol content and enriched in omega-3 FAs and natural antioxidants, providing an alternative for consumers to obtain these valuable health-promoting nutrients. 

Further research needs to be conducted to investigate the impact of these unconventional sources of bioactive compounds on the metabolic and liver health of poultry, along with the gut microbiota and immune system.

## Figures and Tables

**Figure 1 animals-14-00750-f001:**
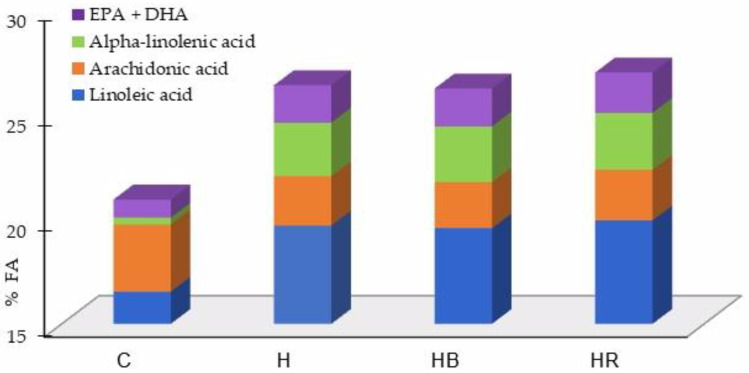
The effects of including hemp seeds alone or in combination with dried fruit pomace on yolk content in major polyunsaturated fatty acids (PUFA) (% of total fatty acids). C: standard diet; H: standard diet containing 8% hemp seed; HB: hemp seed diet containing 3% dried blackcurrant pomace; HR: hemp seed diet containing 3% dried rosehip pomace.

**Figure 2 animals-14-00750-f002:**
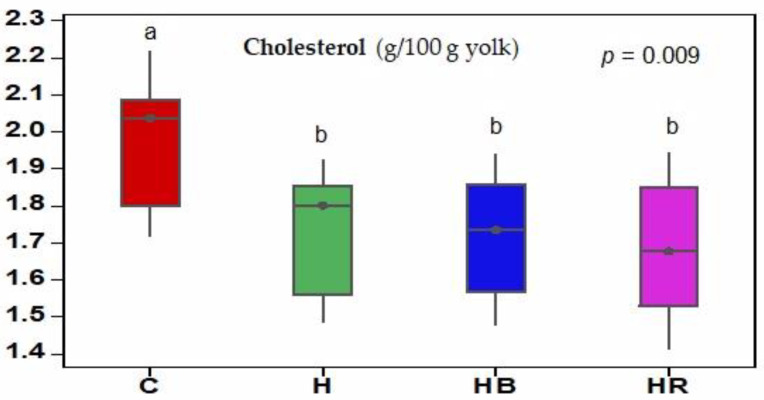
The effects of including hemp seeds alone or in combination with dried fruit pomace on yolk cholesterol content (g/100 g yolk). C: standard diet; H: standard diet containing 8% hemp seed; HB: hemp seed diet containing 3% DB; HR: hemp seed diet containing 3% DR. a, b = between bars, the letter within the same color area differ at *p* < 0.05.

**Figure 3 animals-14-00750-f003:**
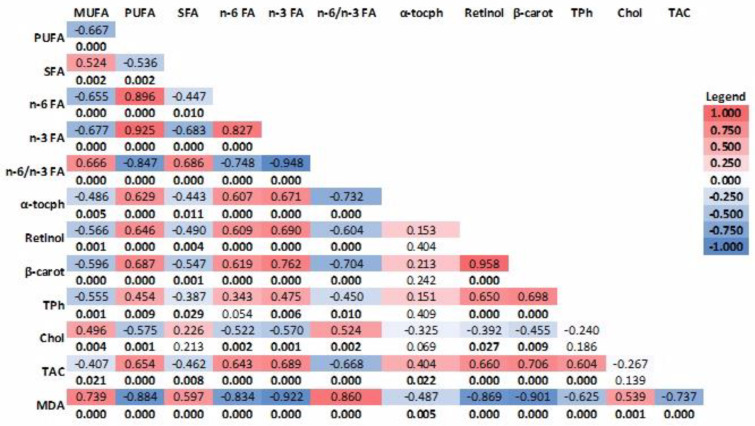
Pearson correlations of FA and antioxidant content and lipid oxidative status of the yolk of stored eggs. The upper value describes the Pearson correlation coefficient (R), and the lower value describes the statistical significance value (*p*-value). The *p*-values in bold face (satisfying the condition *p* < 0.05) emphasize the statistically significant R values. α-tocph: α-tocopherol, β-carot: β-carotene, TPh: total phenols, Chol: cholesterol, TAC: antioxidant capacity.

**Figure 4 animals-14-00750-f004:**
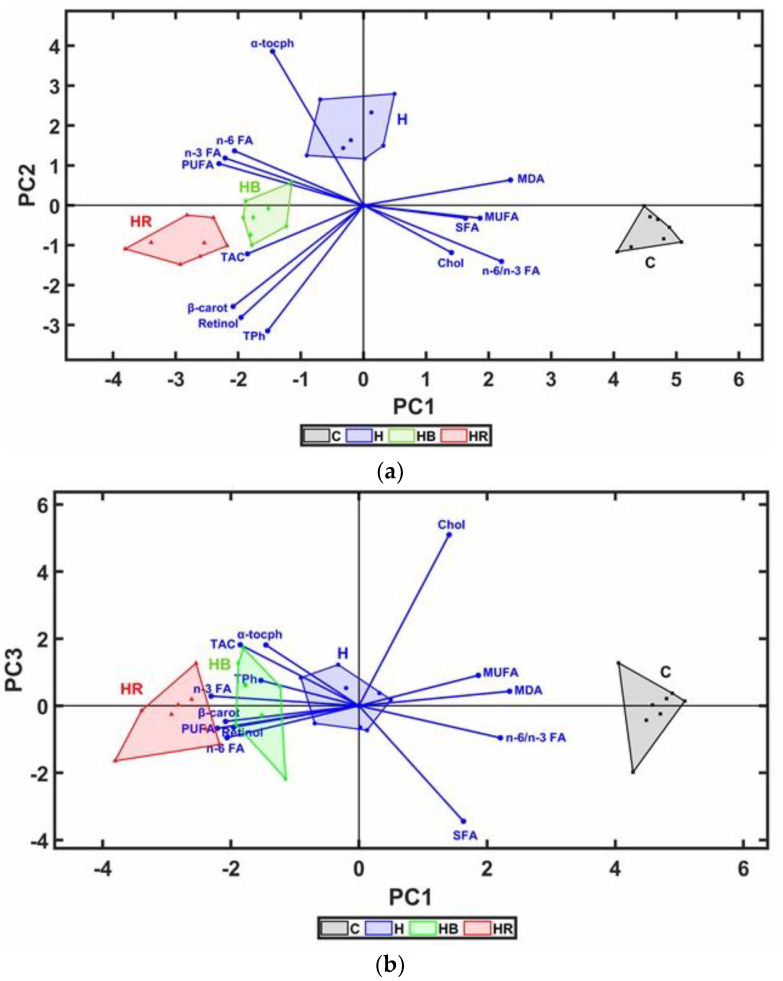
Principal component analysis (PCA, 2D biplot representations) between fatty acids, antioxidants, and egg yolk antioxidant status. (**a**) The first and second discriminant functions; (**b**) the first and third discriminant functions. C: standard diet; H: standard diet containing 8% hemp seed; HB: hemp seed diet containing 3% DB; HR: hemp seed diet containing 3% DR.

**Table 1 animals-14-00750-t001:** Ingredient and chemical composition of the diets used in the study.

Items		Experimental Diets ^1^			
		C	H	HB	HR
Ingredients (% as fed)					
	Corn	44.50	41.35	39.30	38.57
	Wheat	10.00	10.00	10.00	10.00
	Sunflower meal	8.00	6.50	6.20	6.30
	Soybean meal	22.50	20.80	20.18	20.68
	Sunflower oil	3.50	1.85	1.82	1.95
	Hemp seed	-	8.00	8.00	8.00
	Dried blackcurrant pomace (DB)	-	-	3.00	-
	Dried rosehip pomace (DR)	-	-	-	3.00
	Calcium carbonate, amino acids, mineral and vitamin mixture *	11.50	11.50	11.50	11.50
Composition (calculated unless noted)					
	Metabolizable energy (kcal/kg) **	2756	2765	2761	2753
	Crude protein (CP) (analyzed, %)	17.51	17.66	17.48	17.55
	Ether extract (EE) (analyzed, %)	5.32	5.54	5.29	5.60
	NDF (analyzed, %)	8.26	9.81	9.93	10.27
	ADF (analyzed, %)	2.32	3.82	3.68	4.09
	Lysine (%)	0.85	0.85	0.85	0.85
	Methionine (%)	0.42	0.42	0.42	0.42
	Cystine (%)	0.29	0.28	0.28	0.28
	Threonine (%)	0.68	0.68	0.68	0.68
	Tryptophan (%)	0.17	0.17	0.17	0.17
	Calcium (%)	3.90	3.90	3.90	3.90
	Total phosphorus (%)	0.65	0.65	0.65	0.65
	Available phosphorus (%)	0.45	0.45	0.45	0.45
	Sodium (%)	0.15	0.15	0.15	0.15

^1^ C: standard diet; H: standard diet containing 8% hemp seed; HB: hemp seed diet containing 3% dried blackcurrant pomace; HR: hemp seed diet containing 3% dried rosehip pomace. NDF: neutral detergent fiber, ADF: acid detergent fiber. * Calcium carbonate 8.59%, monocalcium phosphate 1.35%, salt 0.3%, DL-Methionine 0.11%, L-lysine 0.08%. Mineral and vitamin mixture provided per kilogram of diet: vit. A, 10,000 IU; vit. D3, 2700 IU; vit. E, 125 IU; vit. K, 5 mg; vit. B12, 0.03 mg; vit. B2, 6.0 mg; vit. B9, 3 mg; vit. B5, 14 mg; niacin, 30 mg; biotin, 0.3 mg; vit. B4, 1.2 mg of choline; 125 mg of antioxidant; 50 mg of Fe; 0.3 mg of Se; 0.5 mg of I; 60 mg of Mn; 15 mg of Cu; 90 mg of Zn. ** Metabolizable energy (kcal/kg) = 2707.71 + 58.63 EE − 16.06 NDF [32].

**Table 2 animals-14-00750-t002:** Proximate composition and antioxidant content of hemp seeds and dried fruit pomace (dried blackcurrant pomace: DB; dried rosehip pomace: DR).

		Hemp Seed	DB	DR	*p*-Value
Proximate composition (% of DM)					
	Dry matter (DM, %)	94.67 ± 1.02	88.24 ± 1.47	92.20 ± 0.89	0.204
	Crude protein (CP)	23.93 ^a^ ± 1.08	11.37 ^b^ ± 1.03	9.23 ^b^ ± 0.59	˂0.001
	Ether extract (EE)	31.86 ^a^ ± 1.54	4.03 ^c^ ± 0.28	12.11 ^b^ ± 0.67	˂0.001
	NDF (neutral detergent fiber)	54.08 ^a^ ± 2.44	32.85 ^b^ ± 2.38	38.73 ^b^ ± 2.12	0.046
	ADF (acid detergent fiber)	31.25 ± 1.88	28.13 ± 1.22	32.46 ± 1.76	0.317
Antioxidant content					
	α-tocopherol (µg/g)	20.05 ^a^ ± 0.97	6.64 ^b^ ± 0.81	22.16 ^a^ ± 1.32	˂0.001
	β-carotene (µg/g)	4.13 ^c^ ± 033	9.39 ^b^ ± 0.54	18.17 ^a^ ± 0.88	˂0.001
	Total phenols (mg GAE/g)	6.41 ^c^ ± 0.33	19.32 ^b^ ± 1.12	24.47 ^a^ ± 1.09	˂0.001

^a–c^ Means within a row with different superscripts are different (*p* < 0.05).

**Table 3 animals-14-00750-t003:** Fatty acid (FA) profile of hemp seed, dried fruit pomace (dried blackcurrant pomace: DB; dried rosehip pomace: DR), and experimental diets.

	Hemp Seed	DB	DR	SEM	*p*-Value	Experimental Diets ^1^				SEM	*p*-Value
						C	H	HB	HR		
Fatty acid composition (% of total FA)											
C16:0	6.24 ^ab^	9.17 ^a^	4.26 ^b^	0.531	0.041	14.75 ^a^	11.09 ^b^	9.21 ^b^	9.63 ^b^	0.372	0.042
C18:0	2.65	1.87	2.75	0.216	0.272	3.58	2.69	3.68	3.07	0.057	0.194
C18:1 cis-9	10.33 ^b^	12.09 ^b^	19.77 ^a^	0.735	0.008	21.23 ^a^	15.01 ^b^	15.87 ^b^	14.01 ^b^	0.195	0.032
C18:2 n-6	55.53 ^a^	45.52 ^b^	48.12 ^ab^	1.249	0.006	51.70 ^a^	39.77 ^b^	38.21 ^b^	40.34 ^b^	0.536	0.007
C18:3 n-3	19.23 ^a^	17.79 ^b^	18.24 ^b^	0.816	0.037	3.16 ^b^	20.05 ^a^	19.45 ^a^	20.69 ^a^	0.231	˂0.001
C22:6 n-3	0.54 ^a^	0.11 ^b^	0.12 ^b^	0.072	˂0.001	0.01 ^b^	0.08 ^a^	0.08 ^a^	0.10 ^a^	0.011	0.044
SFA	10.70 ^b^	16.60 ^a^	9.33 ^b^	0.584	0.025	19.66 ^a^	14.73 ^b^	15.81 ^b^	13.23 ^b^	0.297	0.008
MUFA	10.92 ^b^	13.23 ^b^	21.41 ^a^	0.907	0.008	22.24	19.56	21.43	19.67	0.237	0.159
PUFA	77.72 ^a^	69.55 ^b^	68.31 ^b^	2.012	˂0.001	57.79 ^b^	65.53 ^a^	62.19 ^a^	66.99 ^a^	0.635	0.006
n-6 FA	57.35 ^a^	50.53 ^b^	49.37 ^b^	1.451	0.007	52.94 ^a^	44.03 ^b^	42.04 ^b^	45.01 ^b^	0.491	0.017
n-3 FA	20.37 ^a^	19.02 ^b^	18.94 ^b^	1.346	0.021	4.84 ^b^	21.50 ^a^	20.15 ^a^	21.98 ^a^	0.285	˂0.001
n-6/n-3 FA	2.81 ^a^	2.66 ^b^	2.61 ^b^	0.117	0.039	10.96 ^a^	2.05 ^b^	2.09 ^b^	2.05 ^b^	0.138	˂0.001

^1^ C: standard diet; H: standard diet containing 8% hemp seed; HB: hemp seed diet containing 3% DB; HR: hemp seed diet containing 3% DR. FA: fatty acid; SFA: saturated FA; MUFA: monounsaturated FA; PUFA: polyunsaturated FA. SEM: standard error of the mean. ^a,b^ Means within a row, for the same type of feed, with different superscripts are different (*p* < 0.05).

**Table 4 animals-14-00750-t004:** The effects of including hemp seeds alone or in combination with dried fruit pomace on performance and egg quality of laying hens (average values/treatments).

		Dietary Treatment ^1^				SEM ^2^	*p*-Values
		C	H	HB	HR		
	Hemp seed	-	8.0	8.0	8.0		
	DB	-	-	3.0	-		
	DR	-	-	-	3.0		
Initial weight (g/hen)		1693.5	1678.3	1698.8	1706.3	12.821	0.804
Final weight (g/hen)		1762.6	1745.5	1786.1	1809.5	20.132	0.519
Feed intake (g/hen/d)		113.02	112.87	113.35	114.17	0.512	0.207
Egg production (hen-day, %)		90.62 ^b^	91.48 ^a^	91.51 ^a^	91.88 ^a^	0.523	0.038
Egg weight (g)		60.52 ^b^	61.82 ^a^	61.97 ^a^	62.20 ^a^	0.25	0.009
Egg mass (g/hen/d)		54.84 ^b^	56.55 ^a^	56.71 ^a^	57.15 ^a^	0.44	0.017
Feed conversion ratio (g feed/g egg)		2.061 ^a^	1.995 ^b^	1.998 ^b^	1.997 ^b^	0.011	0.024
Yolk weight (g)		15.32 ^b^	17.25 ^a^	17.38 ^a^	17.34 ^a^	0.15	0.045
Albumen weight (g)		38.79	38.26	38.10	38.32	0.22	0.239
Shell weight (g)		6.40	6.31	6.49	6.54	0.05	0.167
Yolk ratio (%)		28.11	27.90	28.04	27.87	0.37	0.465
Albumen ratio (%)		61.29	61.89	61.48	61.61	0.94	0.129
Shell ratio (%)		10.60	10.21	10.48	10.52	0.08	0.818
Roche yolk color fan score		6.69 ^c^	7.43 ^c^	9.87 ^b^	11.32 ^a^	0.081	˂0.001

DB: dried blackcurrant pomace; DR: dried rosehip pomace; ^1^ C: standard diet; H: standard diet containing 8% hemp seed; HB: hemp seed diet containing 3% DB; HR: hemp seed diet containing 3% DR. ^2^ SEM: standard error of the mean. ^a–c^ Means within a row with different superscripts are different (*p* < 0.05).

**Table 5 animals-14-00750-t005:** The effects of including hemp seeds alone or in combination with dried fruit pomace on FA profile of egg yolks.

				**Dietary Treatment ^1^**				**SEM ^2^**	***p*-Values**
				**C**	**H**	**HB**	**HR**		
			Hemp seed	-	8.0	8.0	8.0		
			DB	-	-	3.0	-		
			DR	-	-	-	3.0		
Total lipids (g/100 g yolk)				29.16	30.84	29.88	30.16	0.298	0.208
Fatty acid (% of total FA)									
		C12:0		0.04	0.04	0.03	0.04	0.011	0.376
		C14:0		0.30	0.26	0.26	0.25	0.017	0.088
		C15:0		0.07	0.06	0.06	0.07	0.002	0.451
		C16:0		24.48 ^a^	22.58 ^b^	22.18 ^b^	22.32 ^b^	0.417	0.039
		C17:0		0.16	0.13	0.12	0.13	0.073	0.074
		C18:0		11.48	11.88	11.73	11.95	0.205	0.196
	Σ saturated FA			36.53 ^a^	34.95 ^b^	34.38 ^b^	34.76 ^b^	0.731	0.048
		C14:1		0.03	0.04	0.03	0.04	0.001	0.509
		C16:1		2.97 ^a^	2.12 ^b^	2.31 ^b^	2.04 ^b^	0.102	0.023
		C17:1		0.10	0.12	0.11	0.12	0.009	0.055
		C18:1 cis-9 (OA)		37.72 ^a^	34.54 ^b^	35.15 ^b^	34.04 ^b^	0.883	0.008
	Σ monounsaturated FA			40.82 ^a^	36.82 ^b^	37.60 ^b^	36.24 ^b^	0.408	0.005
		C18:2 n-6 (LA)		16.53 ^b^	19.67 ^a^	19.54 ^a^	19.91 ^a^	0.763	0.002
		C18:3 n-6		0.11	0.14	0.13	0.13	0.021	0.285
		C20:2 n-6		0.18	0.17	0.17	0.18	0.015	0.502
		C20:3		0.43	0.66	0.59	0.60	0.053	0.127
		C20:4 n-6 (AA)		3.19 ^a^	2.34 ^b^	2.19 ^b^	2.40 ^b^	0.071	0.039
	Σ n-6 PUFA			20.44 ^b^	22.98 ^a^	22.62 ^a^	23.22 ^a^	0.514	0.006
		C18:3 n-3 (ALA)		0.32 ^b^	2.54 ^a^	2.64 ^a^	2.71 ^a^	0.169	˂0.001
		C20:3 n-3		0.27	0.29	0.28	0.29	0.027	0.118
		C20:5 n-3 (EPA)		0.08 ^b^	0.31 ^a^	0.30 ^a^	0.33 ^a^	0.019	˂0.001
		C22:6 n-3 (DHA)		0.78 ^b^	1.47 ^a^	1.50 ^a^	1.58 ^a^	0.056	˂0.001
	Σ n-3 PUFA			1.45 ^b^	4.61 ^a^	4.72 ^a^	4.91 ^a^	0.274	˂0.001
	Σ polyunsaturated FA			21.89 ^b^	27.59 ^a^	27.34 ^a^	28.13 ^a^	0.461	
	Other FA			0.76	0.64	0.68	0.87	0.018	0.426
Σ unsaturated FA				62.71 ^b^	64.41 ^a^	64.94 ^a^	64.37 ^a^	0.386	0.008
Hypercholesterolemic FA ^3^				24.82 ^a^	22.88 ^b^	22.47 ^b^	22.61 ^b^	0.229	0.003
Hypocholesterolemic FA ^4^				59.61 ^b^	62.13 ^a^	62.49 ^a^	62.17 ^a^	0.358	0.005

DB: dried blackcurrant pomace; DR: dried rosehip pomace; ^1^ C: standard diet; H: standard diet containing 8% hemp seed; HB: hemp seed diet containing 3% DB; HR: hemp seed diet containing 3% DR. ^2^ SEM: standard error of the mean. FA: fatty acid; PUFA: polyunsaturated FA; OA: oleic acid; LA: linoleic acid; ALA: α-linolenic acid; AA: arachidonic acid; EPA: eicosapentaenoic acid; DHA: docosahexaenoic acid. ^3^ Hypercholesterolemic FA: (C12:0 + C14:0 + C16:0). ^4^ Hypocholesterolemic FA: (C18:1 + PUFA). ^a,b^ Means within a row with different superscripts are different (*p* < 0.05).

**Table 6 animals-14-00750-t006:** Egg yolk fatty acid profile (% of total FA) before and after storage for 28 days.

		Dietary Treatment ^1^				SEM ^2^	*p*-Values ^3^
		C	H	HB	HR		
	Hemp seed	-	8.0	8.0	8.0		
	DB	-	-	3.0	-		
	DR	-	-	-	3.0		
SFA	Fresh	36.53 ^a^	34.95 ^b^	34.38 ^b^	34.76 ^b^	0.731	0.046
	Stored	38.28 ^b^	36.27 ^a^	35.63 ^a^	35.73 ^a^	0.594	0.021
	SEM ^2^	1.104	0.952	1.212	1.082	-	-
	*p*-values ^4^	0.008	0.087	0.159	0.202	-	-
MUFA	Fresh	40.82 ^a^	36.82 ^b^	37.60 ^b^	36.24 ^b^	0.408	0.007
	Stored	40.31 ^a^	37.52 ^b^	38.03 ^b^	36.35 ^b^	0.529	˂0.001
	SEM ^2^	0.986	0.672	0.815	0.931	-	-
	*p*-values ^4^	0.382	0.291	0.343	0.159	-	-
PUFA	Fresh	21.89 ^b^	27.59 ^a^	27.34 ^a^	28.13 ^a^	0.461	˂0.001
	Stored	17.77 ^c^	25.59 ^b^	25.75 ^b^	27.40 ^a^	0.573	˂0.001
	SEM ^2^	0.712	0.841	0.639	0.511	-	-
	*p*-values ^4^	˂0.001	0.039	0.022	0.174	-	-
n-6 FA	Fresh	20.44 ^b^	22.98 ^a^	22.62 ^a^	23.22 ^a^	0.514	0.003
	Stored	16.55 ^c^	21.73 ^b^	21.30 ^b^	22.97 ^a^	0.378	˂0.001
	SEM ^2^	0.684	0.539	0.894	0.443	-	-
	*p*-values ^4^	0.009	0.042	0.027	0.184	-	-
n-3 FA	Fresh	1.45 ^b^	4.61 ^a^	4.72 ^a^	4.91 ^a^	0.274	˂0.001
	Stored	1.22 ^c^	3.86 ^b^	4.45 ^a^	4.43 ^a^	0.239	˂0.001
	SEM ^2^	0.045	0.059	0.104	0.047	-	-
	*p*-values ^4^	0.006	0.007	0.041	0.037	-	-
n-6/n-3 FA	Fresh	14.10 ^a^	4.98 ^b^	4.79 ^b^	4.73 ^b^	0.216	˂0.001
	Stored	13.56 ^a^	5.63 ^b^	4.78 ^c^	5.18 ^bc^	0.207	˂0.001
	SEM ^2^	0.413	0.302	0.286	0.175	-	-
	*p*-values ^4^	0.064	0.119	0.271	0.092	-	-

DB: dried blackcurrant pomace; DR: dried rosehip pomace; ^1^ C: standard diet; H: standard diet containing 8% hemp seed; HB: hemp seed diet containing 3% DB; HR: hemp seed diet containing 3% DR. ^2^ SEM: standard error of the mean. ^3^ Effect of diets; ^4^ effect of storage. FA: fatty acid; SFA: saturated FA; MUFA: monounsaturated FA; PUFA: polyunsaturated FA. ^a–c^ Means within a row with different superscripts are different (*p* < 0.05).

**Table 7 animals-14-00750-t007:** Estimation of health indices and enzymes of FA metabolism based on the FA composition of egg yolk.

		Dietary Treatment ^1^				SEM ^2^	*p*-Values
		C	H	HB	HR		
	Hemp seed	-	8.0	8.0	8.0		
	DB	-	-	3.0	-		
	DR	-	-	-	3.0		
1. Health indices							
Saturation index (SI)		0.578	0.539	0.526	0.536	0.011	0.537
PUFA/SFA		0.599 ^b^	0.789 ^a^	0.795 ^a^	0.809 ^a^	0.022	0.002
n-6/n-3 FA		14.10 ^a^	4.98 ^b^	4.79 ^b^	4.73 ^b^	0.216	˂0.001
LA/ALA		51.65 ^a^	7.74 ^b^	7.40 ^b^	7.35 ^b^	0.139	˂0.001
Peroxidability index (PoI)		39.47 ^b^	51.00 ^a^	50.49 ^a^	52.68 ^a^	0.673	˂0.001
Oxidative susceptibility (OS)		827.6 ^b^	1189.9 ^a^	1193.9 ^a^	1216.2 ^a^	6.219	˂0.001
Polyunsaturation index (PI)		17.17 ^b^	24.75 ^a^	24.82 ^a^	25.33 ^a^	1.32	˂0.001
Nutritional value index (NVI)		2.01	2.06	2.11	2.06	0.027	0.375
Atherogenicity index (AI)		0.59	0.55	0.54	0.55	0.006	0.098
Thrombogenicity index (TI)		1.03 ^a^	0.79 ^b^	0.78 ^b^	0.77 ^b^	0.032	˂0.001
h/H FA		2.40 ^b^	2.72 ^a^	2.78 ^a^	2.75 ^a^	0.023	˂0.001
Health-promoting index (HPI)		2.43 ^b^	2.72 ^a^	2.79 ^a^	2.76 ^a^	0.032	0.035
Desirable FA (DFA)		74.19 ^b^	76.29 ^a^	76.67 ^a^	76.32 ^a^	0.417	0.007
2. FA metabolism indices							
Elongase index (EI)		0.47 ^b^	0.53 ^a^	0.53 ^a^	0.54 ^a^	0.012	0.004
Thioesterase index (ToI)		81.60 ^c^	86.84 ^b^	85.30 ^b^	89.28 ^a^	3.861	0.009
∆9C18 (DI, 18)		76.66	74.41	74.98	74.02	0.682	0.134
∆9C16 (DI, 16)		10.81	8.58	9.43	8.37	0.318	0.265
∆9-desaturase index (total DI)		53.08	51.55	52.49	51.29	0.873	0.427
∆5/∆6-Desaturase		20.07 ^a^	16.19 ^b^	15.79 ^b^	16.56 ^b^	0.829	0.008

DB: dried blackcurrant pomace; DR: dried rosehip pomace;^1^ C: standard diet; H: standard diet containing 8% hemp seed; HB: hemp seed diet containing 3% DB; HR: hemp seed diet containing 3% DR. ^2^ SEM: standard error of the mean. FA: fatty acid; SFA: saturated FA; PUFA: polyunsaturated FA; LA: linoleic acid; ALA: α-linolenic acid; h/H FA: hypocholesterolemic/hypercholesterolemic ratio. DI: desaturase index. ^a–c^ Means within a row with different superscripts are different (*p* < 0.05).

**Table 8 animals-14-00750-t008:** Egg yolk antioxidant compound concentrations and lipid oxidative status before and after storage for 28 days.

			Dietary Treatment ^1^				SEM ^2^	*p*-Values ^3^
			C	H	HB	HR		
	Hemp seed		-	8.0	8.0	8.0		
	DB		-	-	3.0	-		
	DR		-	-	-	3.0		
1. Antioxidant content of the yolk								
α-tocopherol (µg/g)		Fresh	18.32 ^b^	26.17 ^a^	25.81 ^a^	27.02 ^a^	0.324	0.007
		Stored	13.70 ^b^	18.14 ^a^	16.52 ^ab^	16.18 ^ab^	0.208	0.028
		SEM ^2^	0.441	0.238	0.382	0.573	-	-
		*p*-values ^4^	0.007	0.009	˂0.001	˂0.001	-	-
Retinol		Fresh	30.21 ^b^	31.18 ^b^	35.49 ^a^	38.55 ^a^	0.278	0.005
		Stored	19.73 ^b^	20.67 ^b^	28.57 ^a^	34.59 ^a^	0.832	0.034
		SEM ^2^	1.261	0.892	1.054	0.923	-	-
		*p*-values ^4^	˂0.001	˂0.001	˂0.001	0.007	-	-
β-carotene (µg/g)		Fresh	8.37 ^b^	10.20 ^b^	17.72 ^a^	21.84 ^a^	1.093	˂0.001
		Stored	8.21 ^b^	9.72 ^b^	15.51 ^a^	17.39 ^a^	1.276	˂0.001
		SEM ^2^	1.014	0.538	0.996	1.140	-	-
		*p*-values ^4^	0.313	0.121	0.094	0.072	-	-
Total phenols (mg GAE/g)		Fresh	1.27 ^b^	1.47 ^b^	1.79 ^a^	1.80 ^a^	0.091	0.019
		Stored	1.11 ^b^	1.14 ^b^	1.41 ^a^	1.46 ^a^	0.013	0.042
		SEM ^2^	0.103	0.083	0.124	0.159	-	-
		*p*-values ^4^	0.041	0.024	0.004	0.007	-	-
2. Lipid oxidative status of the yolk								
Antioxidant capacity (µmol TE/g yolk)		Fresh	0.872 ^c^	1.185 ^b^	1.273 ^a^	1.281 ^a^	0.151	0.004
		Stored	0.336 ^c^	0.414 ^b^	0.479 ^a^	0.495 ^a^	0.194	0.031
		SEM ^2^	0.105	0.166	0.044	0.214	-	-
		*p*-values ^4^	˂0.001	˂0.001	˂0.001	˂0.001	-	-
TBARS (µg MDA/g yolk)		Fresh	0.747 ^a^	0.583 ^b^	0.514 ^c^	0.488 ^c^	0.025	0.007
		Stored	1.279 ^a^	1.021 ^b^	0.910 ^c^	0.782 ^d^	0.019	˂0.001
		SEM ^2^	0.061	0.044	0.073	0.026	-	-
		*p*-values ^4^	˂0.001	˂0.001	˂0.001	˂0.001	-	-

DB: dried blackcurrant pomace; DR: dried rosehip pomace; ^1^ C: standard diet; H: standard diet containing 8% hemp seed; HB: hemp seed diet containing 3% DB; HR: hemp seed diet containing 3% DR. ^2^ SEM: standard error of the mean. ^3^ Effect of diets; ^4^ effect of storage. TBARS: thiobarbituric acid reactive substance; MDA: malondialdehyde. ^a–d^ Means within a row with different superscripts are different (*p* < 0.05).

## Data Availability

Data are contained within the article.

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
