# Peer review of "Effect of Dietary Incorporation of Hemp Seeds Alone or with Dried Fruit Pomace on Laying Hens’ Performance and on Lipid Composition and Oxidation Status of Egg Yolks"

_animals, 2024, doi:10.3390/ani14050750_

Round 1
Reviewer 1 Report
Comments and Suggestions for Authors
Please find Comments and Suggestions for Authors in the pdf file attached below.

Author Response
Thank you very much for taking the time to review this manuscript. The authors accepted all comments. Please find the detailed responses below and the corresponding revisions/corrections highlighted/in track changes in the re-submitted files. Your comments helped us to improve the paper. |
Questions for General Evaluation: The topic is of interest since hemp seed have been reported as a source of valuable nutritional compounds and dried fruit pomace are by-product rich in fiber and natural antioxidant and their addition in animal diet is beneficial and sustainable. Conclusion should better emphasize the safe nutritional role of hemp added to the diets of laying hens and the complementary role of dried fruit pomace in increasing the antioxidant content and oxidative stability of fats in yolk. Introduction should be implemented. Material and methods, results are clear and well described.
|
Response and Revisions: Agreed. I, accordingly, modified. Line 727-735: The results of this study provide clear evidence that it is safe to add hemp seeds (8%) to the diets of laying hens and that they can serve as sources of PUFAs for the production of high-quality table eggs, enriched with omega-3 FAs and with smaller cholesterol and SFA content. In addition, a significant enrichment of eggs with α-tocopherol was found which ensured an improvement in the antioxidant status of the fats in the egg yolk. The addition of 3% dried fruit pomace (DB or DR) to the diets of laying hens, which is enriched in PUFAs, can have a positive influence on the parameters of egg quality by increasing the pigmentation and antioxidant content and reducing the MDA concentration in the yolks of fresh or stored eggs (28 days in the refrigerator), demonstrating their efficiency in reducing the oxidation of unsaturated FAs in egg yolks. It can be concluded that including hemp seeds in combination with dried fruit pomace in diets of laying hens did not affect the laying performance of hens, but ensured an improvement in egg quality. This strategy of feeding hens allows to obtain eggs enriched in omega-3 FAs and natural antioxidants, providing an alternative for consumers to obtain these valuable health-promoting nutrients.
|
Comments 1: Line 59-60 Alpha-Linolenic Acid (ALA), eicosapentaenoic acid (EPA) Docosahexaenoic Acid (DHA). In general check all the abbreviations in the manuscript. |
Response 1: Agreed. I, accordingly, modified, following all the abbreviations in the manuscript. Line 59-60: ………. especially alpha-linolenic acid (ALA), eicosapentaenoic acid (EPA) and docosahexaenoic acid (DHA), ……..
|
Comments 2: Line 64-65 remove fish oil and consider only the effect of comparable complex matrix as seeds to enrich eggs n-3 FA. |
Response 2: Agreed. I, accordingly, modified. Line 64-65: Previous reports have shown that table eggs can be enriched with n-3 FA by adding flax seeds [2,6,7], hemp seeds [8,9,10,11], or hempseed cake [11] to the diet. I removed reference 12: Kralik, G,; Kralik, Z.; Grcevic, M.; Galovic, O.; Hanzek, D.; Biazik, E. Fatty acid profile of eggs produced by laying hens fed diets containing different shares of fish oil. Poult. Sci. 2021, 100, 101379.
|
Comments 3: Line 69 consider not only carbohydrates content but also fiber composition. Hemp seed are also rich in functional components that should be mentioned. |
Response 3: Agreed. I, accordingly, modified. Line 69: Whole hemp seeds contain approximately 25% crude protein (CP), 33-35% fat, 34% carbohydrates, crude fiber, vitamins, minerals and functional components [8,11,16].
|
Comments 4: Line 72 please add flaxseed ALA content. |
Response 4: Agreed. I added. Line 72: …………surpassed only by flax seeds (55–57%) [6].
|
Comments 5: Line 73-76 the choice of the content of 8% in hemp seed in the diet even if 30% has no detrimental effect on eggs parameters should be better addressed. |
Response 5: Thank you for pointing this out. I agree with this comment. Your observation is correct and we will take it into account in future studies that we intend to carry out within a project aimed at the sustainable valorization of some agro-industrial by-products as a source of polyphenols in bird feed in relation to the safety and quality of production. The choice of the content of 8% in hemp seed in the diet was based on a previous study (Mierlita, 2019). In addition, the study by Neijat et al. (2014) demonstrated that increasing the content of hemp seed in the diet of hens from 10% to 30% did not alter the laying rate and feed intake.
|
Comments 6: Line 91-92: farm to fork approach should be mentioned |
Response 6: Agreed. I added. Line 96: This sustainable agri-food system promotes from farm to fork strategy which is designed for building a fair, healthy and environmentally-friendly food system.
|
Comments 7: Line 100-104 what is the availability of these products? |
Response 7: In Romania, significant amounts of by-products obtained from the processing of forest fruits result annually. According to INS data (National Institute of Statistics) in 2022 in Romania, over 5, 000 ha were cultivated with forest fruits (blackberries, currants, blueberries, raspberries, strawberries), and over 9,000 tons of forest fruits were harvested from the spontaneous flora and 4,000 tons of rose hips [25].
|
Comments 8: Line 105-112 this paragraph should be reduced and considered for the discussion. In the introduction, authors should address the level of dried fruit pomace considered in the study in line with the literature available. |
Response 8: Agreed. I added. Line 104: In Romania, significant amounts of by-products obtained from the processing of forest fruits result annually. According to INS data (National Institute of Statistics) in 2022 in Romania, over 5, 000 ha were cultivated with forest fruits (blackberries, currants, blueberries, raspberries, strawberries), and over 9,000 tons of forest fruits were harvested from the spontaneous flora and 4,000 tons of rose hips [25]. Reference: 25. INS (National Institute of Statistics) - Vegetable production of the main crops in 2022. https://insse.ro/cms/ro/content/producÈ›ia-vegetală-la-principalele-culturi-în-anul-2022-date-provizorii (accessed 20 February 2024).
|
Comments 9: Line 117 specify in the aim the level of inclusion adopted in the study |
Response 9: Agreed. I added. The aim of the study was to evaluate the effects of including hemp seeds (8%) alone or with dried fruit pomace (DB or DR) (3%) in the diets of layer hens on ……….
|
Comments 10: Table 2 consider also to add ADL content |
Response 10: Thank you for pointing this out. I agree with this comment. The content of ADL in ingredients and diets has not been analyzed in the laboratory.
|
Comments 11: Lines 502-513 Tocopherol content is not mentioned |
Response 11: Agreed. I added. Line 504: The higher concentration of α-tocopherol in DR compared to DB is attributed to the seed content which also provided a higher fat content for DR. Previously Vlaicu et al. [6] reported that rosehip seeds are high content of oil (16.2%) and α-tocopherol (6).
|
Comments 12: Lines 506-508 also the chemical composition of hemp seed is influenced by a number of factors that should be addressed. |
Response 12: Agreed. I added. Line 509: Similarly, the chemical composition of hemp seeds varies depending on different parameters such as environmental conditions, soil composition, fertilizers, and plant variety [15].
|
Comments 13: Lines 511-512 phenol content of hemp seed should be discussed |
Response 13: Agreed. I added. Line 513: The total phenols content of the hemp seeds was lower than that reported in other studies (6.41 vs. 20.19 mg GAE/g) (15). This difference could be attributed to plant variety (different genotypes) and environmental conditions (year of growth) (13). Moreover, it must be taken into account that phenols present a very small portion of the polyphenolic profile and the evaluation of phenols by the Folin-Ciocalteu method always presented lower values than methods that used different solvent mixtures (37).
|
Comments 14: Line 567 detail the role of hemp seed mediated by sitosterol in reducing cholesterol absorption |
Response 14: Agreed. I added. Line 568: Golimowski et al. [54] attributed the decreased cholesterol levels in yolks to the content of ?-sitosterol (phytosterol) in hemp seeds, which has a role in reducing cholesterol absorption through crystallization and coprecipitation. Moreover, ?-sitosterol have lower hydrosolubility than cholesterol so it misplaces the cholesterol from intestinal micelles. This competition reduces the absorption rate of the cholesterol [53].
|
Comments 15: Lines 646-650 mention EFSA as the ideal intake |
Response 15: Agreed. I added. Line 650: The PUFA/SFA and n-6/n-3 PUFA ratios are commonly used to assess the nutritional value of animal fat on consumer health. In general, a ratio of PUFA to SFA above 0.45 and a ratio of n-6/n-3 FA below 4.0 are required in human diets to combat ‘lifestyle diseases’ such as coronary heart disease and cancer [1]. In the present study, the PUFA/SFA ratios (0.59 - 0.80 : 1) were higher than the recommended values for all groups, whereas the n-6/n-3 FA ratios they came closer the recommended levels for the hemp seed supplemented groups (4.73 - 4.98 : 1).
|
Comments 16: Lines 684-688 mention the role of tocopherol in the lipid oxidative process. |
Response 16: Agreed. I added. Line 688: In addition, the lower concentration of MDA in the yolks of HR eggs than HB eggs can be attributed to the higher content of α-tocopherol in HR eggs, which reduced the oxidative process of lipids in the yolk. |

Reviewer 2 Report
Comments and Suggestions for Authors
Dear Author,
Introduction
Lines 54-55: There are different results regarding the negative effects of egg-derived cholesterol on human health. Therefore, it may be more appropriate to use expressions such as "it is claimed, it is thought".
Materials and Methods
Lines 141-146: Please indicate the stocking density as hens/cm2.
Lines 145-146: What criteria did you use to establish the 3% application doses? Kindly provide details in the method section.
Line 201: Delete “strength”
Results
Lines 298-304: Kindly rewrite these statements. The items are compared in these statements, but it's unclear which is different from the other.
Table 2: Seed and fruit pomace are compared in Table 2, but p values and upper letters are not indicated.
Lines -307-313: Kindly rewrite these statements. The items are compared in these statements, but it's unclear which is different from the other.
Table 3: Seed, fruit pomace and diets are compared in Table 3, but p values and upper letters are not indicated.
Line 321: Delete “live”
Line 322: “but it increased by 67.2–103.2 g during the experimental period in all groups of hens”. It would not be appropriate to say this since you did not perform statistical analysis based on age.
Table 4: Replace “laying rate (%)” with “Egg production (hen-day,%)”
Table 4: Replace “ (g/g egg)” with “ (g feed/g egg)
Table 4: Replace “022” with “0.22”
Line 335: “by food treatment.”. Please use an alternative expression for this term.
Table 5: Although the p values of some charecteristics (Hypercholesterolemic FA….) in Table 5 are important, why are they not lettered?
Table 6: It is not clear which factors the p values in Table 6 belong to. It may be more understandable if the P values are written by row and column.
Lines 398-399: It appears that this sentence is incorrect. Please check again.
Table 8: It is not clear which factors the p values in Table 8 belong to. It may be more understandable if the P values are written by row and column.
Discussion
Lines 503-504: Please add the P values to Table 2 and Table 3.
Line 524: Replace “Turkey” with “turkey”
Lines 527-529: Please caheck this sentence. Treatments had an effect on yolk weight.
Line 562: In the results section, the order in the tables is first fatty acid and then cholesterol. It would be more appropriate to do the same order in the discussion section.
Lines 563-564: There are different results regarding the negative effects of egg-derived cholesterol on human health. Therefore, it may be more appropriate to use expressions such as "it is claimed, it is thought".
Lines 578: Replace “Supplementing the diet” with “supplemented with hempseed diet”
Lines 579-580: Please check this references. Researchers used multienzymes in their study. “by Baghban-Kanani et al. [61], who concluded that dietary supplementation with dried and ground rosehips resulted in decreased cholesterol concentration in yolks.
Line : Please check the p value “(p ˃ 0.05)”
Author Response
Thank you very much for taking the time to review this manuscript. The authors accepted all comments. Please find the detailed responses below and the corresponding revisions/corrections highlighted/in track changes in the re-submitted files. Your comments helped us to improve the paper. |
Introduction |
Comments 1: Lines 54-55: There are different results regarding the negative effects of egg-derived cholesterol on human health. Therefore, it may be more appropriate to use expressions such as "it is claimed, it is thought". |
Response 1: Agreed. I, accordingly, modified. Lines 54-55: However, eggs are high in cholesterol and saturated fatty acids (SFAs), which it is claimed contribute to coronary heart disease [1].
|
Materials and Methods
|
Comments 2: Lines 141-146: Please indicate the stocking density as hens/cm2. |
Response 2: Agreed. I added. Lines 148-149: The hens were raised in a shelter equipped with Zucami three-tier metallic cages (60 cm width x 60 cm length x 40 cm height) at a density of 4 hens/cage (900 cm2/hen) …………..
|
Comments 3: Lines 145-146: What criteria did you use to establish the 3% application doses? Kindly provide details in the method section. |
Response 3: Agreed. I added. Line 147: Application doses of 3% for dried fruit pomace were established in accordance with previous studies that demonstrated that the optimal dose of incorporation of sources of natural antioxidants (tomato waste, dehydrated carrots, rosehip meal, dehydrated sea buckthorn pomace, dehydrated kapia peppers) in the diet of laying hens is 2-3% (6, 7). In addition, Konca et al. (30) reported that rose hips can act as a prooxidant at high concentrations of 5% in the diet of laying quail.
|
Comments 4: Line 201: Delete “strength”
|
Response 4: Agreed. I, accordingly, modified Line 201: The yolk color strength was determined using the Roché scale (1–15 points).
|
Results |
Comments 5: Lines 298-304: Kindly rewrite these statements. The items are compared in these statements, but it's unclear which is different from the other. |
Response 5: I agree with this comment. I, accordingly, modified. Line 298: Hemp seeds had higher concentrations of CP and EE (p Ë‚ 0.001), but also a higher concentration of NDF (p Ë‚ 0.05), than dried fruit pomace (Table 2). Dried blackcurrant (DB) and dried rosehip (DR) pomace had higher β-carotene and total phenol content (DR ˃ DB), than hemp seed (p Ë‚ 0.001). The α-tocopherol content was higher in hemp seed and DR than DB (p Ë‚ 0.001).
|
Comments 6: Table 2: Seed and fruit pomace are compared in Table 2, but p values and upper letters are not indicated. |
Response 6: Agreed. I, accordingly, modified. Table 2 has been modified (p-values and upper letters have been added). The p-values have also been added in the text of the manuscript.
|
Comments 7: Lines -307-313: Kindly rewrite these statements. The items are compared in these statements, but it's unclear which is different from the other. |
Response 7: Agreed. I added. Line 307: The FA profile showed higher concentrations of oleic acid (OA, C18:1 c9) (p Ë‚ 0.01) and an n-6/n-3 ratio smaller (p Ë‚ 0.05) in dried fruit pomace, than in hemp seed (Table 3). The hemp seeds had higher concentrations of LA (C18:2n-6) (p Ë‚ 0.01) and ALA (C18:3n-3) (p Ë‚ 0.05) than dried fruit pomace.
|
Comments 8: Table 3: Seed, fruit pomace and diets are compared in Table 3, but p values and upper letters are not indicated. |
Response 8: Agreed. I, accordingly, modified. Table 3 has been modified (p-values and upper letters have been added). The p-values have also been added in the text of the manuscript.
|
Comments 9: Line 321: Delete “live” |
Response 9: Agreed. Line 321: The live body weight (BW) of the laying hens…….
|
Comments 10: Line 322: “but it increased by 67.2–103.2 g during the experimental period in all groups of hens”. It would not be appropriate to say this since you did not perform statistical analysis based on age. |
Response 10: Agreed. The comment is correct. Consequently we have removed this statement. The live body weight (BW) of the laying hens was not affected by the experimental diets, but it increased by 67.2–103.2 g during the experimental period in all groups of hens (Table 4).
|
Comments 11: Table 4: Replace “laying rate (%)” with “Egg production (hen-day,%)” |
Response 11: Agreed. I, accordingly, modified.
|
Comments 12: Table 4: Replace “ (g/g egg)” with “ (g feed/g egg) |
Response 12: Agreed. I, accordingly, modified.
|
Comments 13: Table 4: Replace “022” with “0.22” |
Response 13: Agreed. I, accordingly, modified.
|
Comments 14: Line 335: “by food treatment.”. Please use an alternative expression for this term. |
Response 14: Agreed. I, accordingly, modified. Line 335: The weight of egg albumen and mineral peel was not influenced (p > 0.05) by experimental diets.
|
Comments 15: Table 5: Although the p values of some characteristics (Hypercholesterolemic FA….) in Table 5 are important, why are they not lettered? |
Response 15: Thanks for the comment. We agree. We omitted to write the letters. We corrected and lettered for p ≤ 0.05 values (Σ unsaturated FA, Hypercholesterolemic FA and Hypocholesterolemic FA) in Table 5. We checked this comment for all tables.
|
Comments 16: Table 6: It is not clear which factors the p values in Table 6 belong to. It may be more understandable if the P values are written by row and column. |
Response 16: Agreed. I added. Table 6 has been modified. p-values were written per column (for the effect of diets) and per row (for the effect of storage). The p-values have also been added in the text of the manuscript. Thanks for the suggestion.
|
Comments 17: Lines 398-399: It appears that this sentence is incorrect. Please check again. |
Response 17: Thanks for the comment. We agree. Lines: 398-399. The sentence you refer to is incorrect, because the differences were not statistically ensured (p ˃ 0.05). Consequently, we removed this sentence from the manuscript.
|
Comments 18: Table 8: It is not clear which factors the p values in Table 8 belong to. It may be more understandable if the P values are written by row and column. |
Response 18: Agreed. I added. Table 8 has been modified. p-values were written per column (for the effect of diets) and per row (for the effect of storage). The p-values have also been added in the text of the manuscript. Thanks for the suggestion.
|
Discussion |
Comments 19: Lines 503-504: Please add the P values to Table 2 and Table 3. |
Response 19: Agreed. I added. Lines 503-504: As expected, hemp seeds had higher protein and fat concentrations (p Ë‚ 0.001) than dried fruit pomace (DB and DR), which instead were a good source of β-carotene and total phenols (p Ë‚ 0.001).
|
Comments 20: Line 524: Replace “Turkey” with “turkey” |
Response 20: Agreed. I, accordingly, modified. |
Comments 21: Lines 527-529: Please check this sentence. Treatments had an effect on yolk weight. |
Response 21: Agreed. I, accordingly, modified. Line 527: Supplementing the diets with hemp seeds did not affect the weight and the proportion of egg white, which agrees with the conclusions of Skřivan et al. [43] but not with those of Halle and Schone [45], who observed a higher proportion of egg white.
|
Comments 22: Line 562: In the results section, the order in the tables is first fatty acid and then cholesterol. It would be more appropriate to do the same order in the discussion section. |
Response 22: Thanks for the observation. The comment is correct, accordingly I have modified, the section on cholesterol has been moved after fatty acids.
|
Comments 23: Lines 563-564: There are different results regarding the negative effects of egg-derived cholesterol on human health. Therefore, it may be more appropriate to use expressions such as "it is claimed, it is thought". |
Response 23: Agreed. I, accordingly, modified. Lines 563-564: Due to high cholesterol content, it is thought eggs are with adverse effects on human health [52].
|
Comments 24: Lines 578: Replace “Supplementing the diet” with “supplemented with hempseed diet” |
Response 24: Agreed. I, accordingly, modified.
|
Comments 25: Lines 579-580: Please check this references. Researchers used multienzymes in their study “by Baghban-Kanani et al. [61], who concluded that dietary supplementation with dried and ground rosehips resulted in decreased cholesterol concentration in yolks. |
Response 25: Thank you for pointing this out. I agree with this comment. I modified. Lines 579 - 580: Supplementing the diet with dried fruit pomace (DB or DR) did not cause decreased egg yolk cholesterol levels, according to the results reported by Grigorova et al. [27], who concluded that dietary supplementation with 0.5% dried and milled fruits of rosehip did not change the cholesterol concentration in the yolk.
|
Comments 26: Line : Please check the p value “(p ˃ 0.05)” |
Response 26: Agreed. I, accordingly, modified. Lines 614-617: As differences in yolk ALA and DHA content were not statistically significant (p ˃ 0.05), we removed this paragraph from the manuscript. |

Reviewer 3 Report
Comments and Suggestions for Authors
The search for additives or alternative ingredients for the diet of laying hens with eggs in mind has driven a lot of research; with the appeal of the nutraceutical effect. Congratulations to the authors for the work, with interesting results, which can be published. I will make comments on the text below, suggesting adjustments before publication in order to improve the text and its understanding.
1) throughout the text the authors used the term "supplementary / supplementation", but this term is incorrect. Adjust all text. This supplementary term is to provide something in the diet "with an additional dose than what you are already eating normally". For minerals, for example, this supplementary term is correct; but here it is wrong.
2) The conclusion at the end of the summary section; must respond to the objective.
3) The introduction was long; but like a lot of relevant information; however, this section was left without the authors' hypothesis for the experiment. What was expected?
4) Congratulations on the material and methods section, very well described; with detailed methodologies. The data analysis was also described clearly and I believe it to be adequate.
5) In the results section, I suggest that every time the authors mention that a result is significant, add the P value. This was done in a large part of the section, however other parts were left out. review.
6) Was data from table 2 statistically analyzed? make this clear in the footer of this table; as well as the others.
7) I didn't understand the data in figure 1. It needs to be made clear.
8) In the other figures, I also had difficulty; It needs to be very clear in the table caption what readers need to see.
9) I believe the discussion section was more difficult to write; as there are few studies on the subject; but I believe that the discussion of the mechanisms involved in the changes presented can be further discussed.
10) good conclusion; showing the new data that this work brought.
11) Pay attention to the formatting of references.
I hope you enjoy the tips to improve your paper.
Author Response
Thank you very much for taking the time to review this manuscript. The authors accepted all comments. Please find the detailed responses below and the corresponding revisions/corrections highlighted/in track changes in the re-submitted files. Your comments helped us to improve the paper. |
Comments 1: throughout the text the authors used the term "supplementary / supplementation", but this term is incorrect. Adjust all text. This supplementary term is to provide something in the diet "with an additional dose than what you are already eating normally". For minerals, for example, this supplementary term is correct; but here it is wrong.
|
Response 1: Agreed. The comment is correct and accordingly, I modified. We replaced the term "supplementary / supplementation" with: incorporation / introduction / used / diets containing
The incorporation of hemp seed or dried fruit pomace in diets of laying hens ……….. The introduction of hemp seed or dried fruit pomace in diets of laying hens ……… Hemp seed and dried fruit pomace used together in laying hens diet …… Laying hens fed diets containing hemp seed and dried fruit pomace ……. Dietary incorporation of hemp seeds………
|
Comments 2: The conclusion at the end of the summary section; must respond to the objective. |
Response 2: Agreed. I, accordingly, added. Lines 24-27: It can be concluded that including hemp seeds in combination with dried fruit pomace in diets of laying hens did not affect the laying performance of hens, but ensured an improvement in egg quality. This strategy of feeding hens allows to obtain eggs enriched in omega-3 FAs and natural antioxidants, providing an alternative for consumers to obtain these valuable health-promoting nutrients. |
Comments 3: The introduction was long; but like a lot of relevant information; however, this section was left without the authors' hypothesis for the experiment. What was expected? |
Response 3: Agreed. I, accordingly, added. Line 121: In this experiment we evaluated the hypothesis that the simultaneous inclusion of hemp seeds and dried fruit pomace in diets for laying hens increases the concentration of PUFA and natural antioxidants in the yolk and improves oxidative stability of the egg yolk, without affecting egg production.
|
Comments 4: Congratulations on the material and methods section, very well described; with detailed methodologies. The data analysis was also described clearly and I believe it to be adequate. |
Response 4: Thanks for the appreciations.
|
Comments 5: In the results section, I suggest that every time the authors mention that a result is significant, add the P value. This was done in a large part of the section, however other parts were left out. review. |
Response 5: Agreed. I, accordingly, added. I revised the Results section and the Discussion section.
|
Comments 6: Was data from table 2 statistically analyzed? make this clear in the footer of this table; as well as the others. |
Response 6: Agreed. I, accordingly, added. Line 306: Data in Table 2 and Table 3 were statistically analyzed and p-values were reported. Table 2 and Table 3 were modified in the manuscript. p-values were also reported in the text.
|
Comments 7: I didn't understand the data in figure 1. It needs to be made clear. |
Response 7: Agreed. I, accordingly, added. I have improved figure 1 so that it is easier to understand. I replaced in the manuscript.
|
Comments 8: In the other figures, I also had difficulty; It needs to be very clear in the table caption what readers need to see. |
Response 8: Agreed. I, accordingly, added. We have improved the presentation and description for figure 2 and figure 4.
|
Comments 9: I believe the discussion section was more difficult to write; as there are few studies on the subject; but I believe that the discussion of the mechanisms involved in the changes presented can be further discussed. |
Response 9: Agreed. The Discussion section was improved upon the recommendation of the other reviewers regarding the mechanisms by which hemp seed decreased the cholesterol content of the yolk and how the dried fruit pomace increased the oxidative stability of the yolk lipids.
|
Comments 11: good conclusion; showing the new data that this work brought. |
Response 11: Thanks for the appreciation.
|
Comments 12: Pay attention to the formatting of references. |
Response 12: Agreed. Revised the formatting of the references. |

Reviewer 4 Report
Comments and Suggestions for Authors
The manuscript aimed to evaluate the impact of dietary utilisation of hemp seed and dried fruit pomace on laying hens’ performance and on the lipid composition and oxidation status of egg yolks. The authors addressed the issue of enriching eggs with polyunsaturated fatty acids, which may increase susceptibility to lipid peroxidation in the yolks, potentially affecting the nutritional and sensory quality of the eggs and compromising consumer safety.
In general, the manuscript is written in a clear and understandable manner. The manuscript is well-organized, providing an overview of the research background and main objectives, along with a thorough presentation of the methodology. The research findings are presented with clarity through tables and graphs, and a thorough discussion and comparison with relevant recent literature have been conducted. The presented results hold significant importance in the field of poultry nutrition, providing novel insights into the use of alternative sources of polyunsaturated fatty acids and antioxidants to preserve egg quality during storage which could have a beneficial impact on both the industry and consumers.
More specific comments are listed as follows:
P6, L229: provide information on the film thickness for the HP-88 column.
P10, L335: mineral peel? Use appropriate term
P19, L614-615: Your statement “Hens that received feed supplemented with 3% dried fruit pomace (HB and HR) tended to produce eggs with higher ALA and DHA content (p ˃ 0.05)” is contradictory to the results presented in Table 5. According to Table 5, the content of ALA in HB and HR groups was significantly higher than that in C group.
Fatty acid metabolism indices presented in Table 7 are not discussed in the Discussion section.
P21, Conclusions section: The effect of the addition of hempseed and dried fruit pomace on the cholesterol level of egg yolks is not mentioned in this section.
Generally, the presentation of the impact of hempseed and dried fruit pomace on the cholesterol level in egg yolks is contradictory throughout the entire study. Specifically, based on Figure 2, yolk cholesterol levels are significantly reduced in H, HB and HR groups compared to C group. On the other hand, the authors, in certain parts of the paper, state that the yolk cholesterol level was not influenced by the diet. For example, in L21-22, “Dietary incorporation of hemp seeds improved the fat quality of egg yolks by reducing the concentration of cholesterol and SFAs and increasing the proportion of omega-3 FAs. Laying performance, cholesterol concentration, and yolk FA profile were not improved by supplementing with dried fruit pomace….” Additionally, L37-38 “The yolks of H, HB, and HR eggs had lower cholesterol (p Ë‚ 0.01) and SFA content….”; “The introduction of dried fruit pomace (DB or DR) into the diets had no effect on the laying performance of the hens or the cholesterol content and FA profile of the egg yolks…” (L43-45). Likewise, in L578-579 “Supplementing the diet with dried fruit pomace (DB or DR) did not cause decreased egg yolk cholesterol levels…” The authors are advised to review their results and enhance the presentation of the obtained findings and discussion.
Author Response
Thank you very much for taking the time to review this manuscript. The authors accepted all comments. Please find the detailed responses below and the corresponding revisions/corrections highlighted/in track changes in the re-submitted files. Your comments helped us to improve the paper. |
Comments 1: L229: provide information on the film thickness for the HP-88 column. |
Response 1: Agreed. I added. Line 229: (100 m long, 0.25 mm diameter, and 0.20 μm film thickness) |
Comments 2: L335: mineral peel? Use appropriate term |
Response 2: Agreed. The weight of egg albumen and shell was not influenced ….. |
Comments 3: L614-615: Your statement “Hens that received feed supplemented with 3% dried fruit pomace (HB and HR) tended to produce eggs with higher ALA and DHA content (p ˃ 0.05)” is contradictory to the results presented in Table 5. According to Table 5, the content of ALA in HB and HR groups was significantly higher than that in C group. |
Response 3: Agreed. Your observation is correct if the comparison is made with the control diet (C). To eliminate confusion, the sentence has been reworded:
Line 614-615: Hens that received feed supplemented with 3% dried fruit pomace (HB and HR) tended to produce eggs with higher ALA and DHA content (p ˃ 0.05), compared to hens fed the diet supplemented only with hemp seeds (HB and HR vs. H).
Diets H, HB and HR: all contained hemp seeds - the comparison was between the diet that contained only hemp seeds (H) and the diets that contained hemp seeds + dried fruit pomace (HB and HR). The control diet (C) contained neither hemp seeds nor dried fruit pomace.
This paragraph was removed from the manuscript, at the suggestion of a reviewer, because the differences are not statistically assured (p ˃ 0.05).
|
Comments 4: Fatty acid metabolism indices presented in Table 7 are not discussed in the Discussion section. |
Response 4: Agreed. I added. Line 676: From the fatty acid metabolism indices group of indices elongase was significantly higher (p = 0.004) in H, HB and HR eggs compared with C, while thioesterase was significant higher (p = 0.009) in HR compared with C, H and HB. Similar results were obtained by Vlaicu et al. [6] when the hens' diet was supplemented with faxseed meal in combination with rosehip meal. It should be noted that these indexes have low relevance for discriminating eggs quality because they are not able to discriminate metabolic changes due to dietary effects [15]. No effect was found for ∆9-desaturase, that catalyzes the conversion of C16:0 and C18:0 to C16:1 and C18:1, suggesting that the differently concentrations of C16:1 and C18:1 in eggs it is not due ∆9-desaturase activity. The ∆5/∆6-desaturase complex represents the most valid tool to verify the capacity of animals to synthesize LC-PUFAs (long-chain PUFAs) from precursors. The lower ∆5/∆6-desaturase index in H, HB and HR eggs than in C (p ˂ 0.01) demonstrates competition between n-6 and n-3 FA substrate in the desaturation and elongation pathway. In agreement with the results of this study, Mazalli et al. [55] demonstrated that diets containing higher amounts of ALA (alpha-linolenic acid) increase the activity of ∆6-desaturase and increase in long-chain 22-carbon fatty acids, which may be attributed to the use of ALA as the preferred substrate over LA.
|
Comments 5: Conclusions section: The effect of the addition of hempseed and dried fruit pomace on the cholesterol level of egg yolks is not mentioned in this section. |
Response 5: Agreed. I, accordingly, modified. The results of this study provide clear evidence that it is safe to add hemp seeds (8%) to the diets of laying hens and that they can serve as source of PUFAs for the production of high-quality table eggs, enriched with omega-3 FAs and with smaller cholesterol and SFA content. In addition, a significant enrichment of eggs with α-tocopherol was found which ensured an improvement in the antioxidant status of the fats in the egg yolk. The addition of 3% dried fruit pomace (DB or DR) to the diets of laying hens, which is enriched in PUFAs, can have a positive influence on the parameters of egg quality by increasing the pigmentation and antioxidant content and reducing the MDA concentration in the yolks of fresh or stored eggs (28 days in the refrigerator), demonstrating their efficiency in reducing the oxidation of unsaturated FAs in egg yolks. It can be concluded that including hemp seeds in combination with dried fruit pomace in diets of laying hens did not affect the laying performance of hens, but ensured an improvement in egg quality. This strategy of feeding hens allows to obtain eggs with a lower cholesterol content and enriched in omega-3 FAs and natural antioxidants, providing an alternative for consumers to obtain these valuable health-promoting nutrients.
|
Comments 6: Generally, the presentation of the impact of hempseed and dried fruit pomace on the cholesterol level in egg yolks is contradictory throughout the entire study. Specifically, based on Figure 2, yolk cholesterol levels are significantly reduced in H, HB and HR groups compared to C group. On the other hand, the authors, in certain parts of the paper, state that the yolk cholesterol level was not influenced by the diet. For example, in L21-22, “Dietary incorporation of hemp seeds improved the fat quality of egg yolks by reducing the concentration of cholesterol and SFAs and increasing the proportion of omega-3 FAs. Laying performance, cholesterol concentration, and yolk FA profile were not improved by supplementing with dried fruit pomace….” Additionally, L37-38 “The yolks of H, HB, and HR eggs had lower cholesterol (p Ë‚ 0.01) and SFA content….”; “The introduction of dried fruit pomace (DB or DR) into the diets had no effect on the laying performance of the hens or the cholesterol content and FA profile of the egg yolks…” (L43-45). Likewise, in L578-579 “Supplementing the diet with dried fruit pomace (DB or DR) did not cause decreased egg yolk cholesterol levels…” The authors are advised to review their results and enhance the presentation of the obtained findings and discussion. |
Response 6: Agreed. I, accordingly, modified.
Some sentences have been reworded to eliminate confusion. Diets that were supplemented with dried fruit pomace were compared to diets that were supplemented only with hemp seeds. Diets with dried fruit pomace contained: hemp seed + dried fruit pomace (HB and HR), one diet contained only hemp seed (H) and one control diet (C); the comparisons were: C vs. H, HB, HR; H vs. HB, HR and HB vs. HR.
Line 21-22: Laying performance, cholesterol concentration, and yolk FA profile were not improved by supplementing diets with dried fruit pomace, compared to the diet supplemented only with hemp seeds. The incorporation of dried fruit pomace in diets it increased the antioxidant content and oxidative stability of fats in yolks, and improved the color of the yolk.
Line 43-45: The introduction of dried fruit pomace (DB or DR) into the diets had no effect on the laying performance of the hens or the cholesterol content and FA profile of the egg yolks, compared to the diet supplemented only with hemp seeds. The dried fruit pomace it improved the color, accumulation of antioxidants, and oxidative stability of fats in the yolks of fresh eggs and eggs stored at 4 °C for 28 days.
Line 578-579: The incorporation of hemp seed or dried fruit pomace in diets of laying hens did not cause decreased egg yolk cholesterol levels, compared to the diet supplemented only with hemp seeds. Similar results were reported by Grigorova et al. [27], who concluded that dietary supplementation with 0.5% dried and milled fruits of rosehip did not change the cholesterol concentration in the yolk.
|

Round 2
Reviewer 2 Report
Comments and Suggestions for Authors
Dear Authors,
The identified issues have been corrected.
Best regards,
Reviewer 3 Report
Comments and Suggestions for Authors
Very pleased with the review of the manuscript; the adjustments made a huge improvement to the text; a refinement that allowed understanding the relationship between all the variables evaluated; as well as the analysis of this data. Congratulations.